# Continuous Simplicial Neural Networks

**Aref Einizade**
LTCI, Télécom Paris
Institut Polytechnique de Paris
aref.einizade@telecom-paris.fr

**Dorina Thanou**
EPFL, Lausanne, Switzerland
dorina.thanou@epfl.ch

**Fragkiskos D. Malliaros**
CentraleSupélec, Inria
Université Paris-Saclay
fragkiskos.malliaros@centralesupelec.fr

**Jhony H. Giraldo**
LTCI, Télécom Paris
Institut Polytechnique de Paris
jhony.giraldo@telecom-paris.fr

## Abstract

Simplicial complexes provide a powerful framework for modeling higher-order interactions in structured data, making them particularly suitable for applications such as trajectory prediction and mesh processing. However, existing simplicial neural networks (SNNs), whether convolutional or attention-based, rely primarily on discrete filtering techniques, which can be restrictive. In contrast, partial differential equations (PDEs) on simplicial complexes offer a principled approach to capture continuous dynamics in such structures. In this work, we introduce **co**ntinuous **sim**plicial neural netw**o**rk (COSIMO), a novel SNN architecture derived from PDEs on simplicial complexes. We provide theoretical and experimental justifications of COSIMO's stability under simplicial perturbations. Furthermore, we investigate the over-smoothing phenomenon—a common issue in geometric deep learning—demonstrating that COSIMO offers better control over this effect than discrete SNNs. Our experiments on real-world datasets demonstrate that COSIMO achieves competitive performance compared to state-of-the-art SNNs in complex and noisy environments. The implementation codes are available in https://github.com/ArefEinizade2/COSIMO.

## 1 Introduction

Graph representation learning provides a powerful framework for modeling structured data. In this context, graph neural networks (GNNs) have gained significant attention [1, 2, 3, 4], extending neural network architectures to graph-structured data. By capturing complex relationships between nodes, GNNs have been successfully applied to various domains, including computational/digital pathology [5], social network analysis [6], drug discovery [7], materials modeling [8], and computer vision [9, 10]. However, traditional GNNs primarily focus on pairwise interactions between nodes, limiting their ability to model higher-order relationships in complex systems such as biological networks [11]. To overcome this limitation, researchers have explored more expressive mathematical structures, such as *abstract simplicial complexes* [12], generalizing graphs by incorporating multi-way connections and their Laplacians by introducing the Hodge decomposition theory [12, 13].

An abstract simplicial complex is a combinatorial structure composed of sets that are closed under subset operations. For instance, a three-dimensional simplicial complex includes tetrahedrons (four-element sets), triangles (three-element sets), edges (two-element sets), and vertices (one-element sets), as illustrated in Fig. 1. Graphs correspond to simplicial 1-complexes, containing only nodes and edges, while point clouds (sets of unconnected nodes) can be seen as simplicial 0-complexes.

39th Conference on Neural Information Processing Systems (NeurIPS 2025).

Building upon the mathematical foundation of abstract simplicial complexes, simplicial neural networks (SNNs) [14] have emerged as a powerful approach for learning on higher-order structures. Most existing SNNs rely on discrete simplicial filters [14, 15] and their variations [16, 17, 18] to process data defined on simplicial complexes. However, a fundamental question remains largely unexplored: *how can we design continuous SNNs?* Continuous models play a crucial role in capturing real-world dynamics evolving on structured data [19]. Compared to their discrete counterparts, continuous convolutional models offer several advantages, including: *i*) better control of over-smoothing [19], preventing excessive feature homogenization across the structure, and *ii*) greater robustness to structural perturbations [20, 21]. Despite these benefits, to the best of our knowledge, no continuous filtering method has been proposed for learning on simplicial complexes. To address this gap, we introduce the **co**ntinuous **sim**plicial neural netw**o**rk (COSIMO) model, the first method for modeling and learning over the continuous dynamics of simplicial complexes with higher-order connections. We analyze COSIMO both theoretically and empirically, demonstrating its effectiveness in learning from simplicial complex data. Our main contributions can be summarized as follows:

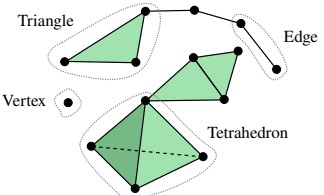

Figure 1: Example of an abstract simplicial complex.

- We introduce COSIMO, a new approach that uses partial differential equations (PDEs) on simplicial complexes to enable continuous information flow through these structures.
- We establish theoretical stability guarantees for COSIMO, demonstrating its robustness to structural perturbations.
- We provide a detailed analysis of over-smoothing in both discrete and continuous SNNs, showing that COSIMO achieves a better control on the rate of convergence to the over-smoothing state.
- We validate COSIMO through experiments on both synthetic and real-world datasets, showing its competitive performance against state-of-the-art (SOTA) methods.

## 2 Related Work

The introduction of topological signal processing over simplicial complexes [22, 13] has significantly advanced topological methods in machine learning, highlighting the benefits of these data structures [12], and contributing to the emerging field of topological deep learning [23, 24]. The evolution of the SNN architectures has followed a trajectory similar to that of GNNs, with the following stages: *i*) the establishment of principles in *topological signal processing over simplicial complexes* [22], *ii*) the formulation of *simplicial filters* [12], and *iii*) the application of these filters to create *SNNs* [14].

Research on learning methods for simplicial complexes has explored various approaches. Roddenberry and Segarra [25] were among the first to develop neural networks that operate on the graph edges using simplicial representations. Building on this, Ebli *et al.* [14] incorporated the Hodge Laplacian, a generalization of the graph Laplacian, to extend GNNs to higher-dimensional simplices. Other studies [26, 17] refined this approach by separating the lower and upper Laplacians, two components of the Hodge Laplacian that capture connections between simplices of different dimensions. Keros *et al.* [27] further extended the framework in [26] to detect topological holes, while Chen *et al.* [28] integrated node and edge operations for link prediction. More recently, attention mechanisms have been incorporated into simplicial networks [29, 11, 30].

Most of these studies focus on learning within individual simplicial levels without explicitly modeling the incidence relations (interactions between simplices of different dimensions) inherent in simplicial complexes [18]. The inclusion of these relations was explored in [18, 17, 31], which proposed convolutional-like architectures that were later unified under the simplicial complex convolutional neural network (SCCNN) framework [18]. Simultaneously, other studies extended message-passing from GNNs [32] to simplicial complexes by leveraging both adjacency and incidence structures [16, 33].

Unlike GNNs, the theoretical understanding of SNNs is still developing. For example, Roddenberry *et al.* [26] analyzed how permutation-symmetric neural networks preserve equivariance under permutation and orientation changes—an important property also supported by SCCNNs [18]. Another study examined message-passing in simplicial complexes through the lens of the Weisfeiler-Lehman test, applied to simplicial complexes derived from clique expansions of graphs [16]. A

spectral formulation based on the simplicial complex Fourier transform was also proposed in [17]. Other critical yet underexplored theoretical aspects of SNNs, including their stability and over-smoothing behaviors, have been analyzed only in the context of SCCNNs [18]. For example, Yang *et al.* [18] analyzed the steady state solution of the diffusion PDE on simplicial complexes, showing that SCCNNs are more robust to over-smoothing compared to their non-Hodge-aware counterparts.

In contrast to previous methods, COSIMO leverages continuous dynamics in both the lower and upper Hodge Laplacians. Although the discrete SCCNN also decouples the Hodge Laplacians, it faces two challenges: *i)* the Hodge filters' order must be manually tuned, and *ii)* the model has limited control over over-smoothing, a common issue in deep GNNs and SNNs. COSIMO addresses these challenges by introducing lower and upper Hodge-aware PDEs on simplicial complexes, enabling differentiability of the convolutional operation *w.r.t.* the simplicial receptive fields, providing greater flexibility and robustness in learning.

## 3 Preliminaries

### 3.1 Notation and Simplicial Complexes

**Notation.** Calligraphic letters like $\mathcal{X}$ denote sets with $|\mathcal{X}|$ being their cardinality. Uppercase boldface letters such as $\mathbf{B}$ represent matrices, and lowercase boldface letters like $\mathbf{x}$ denote vectors. Similarly, $\mathrm{tr}(\cdot)$ represents the trace of a matrix, $\|\cdot\|$ is the $\ell_2$-norm of a vector, and $(\cdot)^\top$ designates transposition.

**Simplicial complex.** A simplicial complex is a set $\mathcal{X}$ of finite subsets of another set $\mathcal{V}$ that is closed under restriction, *i.e.*, $\forall\, s^k \in \mathcal{X}$, if $s^{k'} \subseteq s^k$, then $s^{k'} \in \mathcal{X}$. Each element of $\mathcal{X}$ is called a *simplex*. Particularly, if $|s^k| = k+1$, we call $s^k$ a $k$-simplex. A *face* of $s^k$ is a subset with cardinality $k$, while a *coface* of $s^k$ is a $k+1$-simplex that has $s^k$ as a face [22]. We refer to the 0-simplices as nodes, the 1-simplices as edges, and the 2-simplices as triangles. For higher-order simplices, we use the term $k$-simplices. The notation $\mathcal{X}_k$ represents the collection of $k$-simplices of $\mathcal{X}$. If $\mathcal{X}_c = \emptyset \,\forall\, c > d$, we say $\mathcal{X}$ is a simplicial complex of dimension $d$. For example, a simple graph is a simplicial complex of dimension one and can be represented as $G = (\mathcal{X}_0, \mathcal{X}_1)$, *i.e.*, the set of nodes and edges.

We use incidence matrices $\mathbf{B}_k \in \{-1, 0, 1\}^{|\mathcal{X}_{k-1}| \times |\mathcal{X}_k|}$, to describe the incidence relationships between $k-1$-simplices (faces) and $k$-simplices. For example, $\mathbf{B}_1$ and $\mathbf{B}_2$ are node-to-edge and edge-to-triangle incidence matrices, respectively. Simplicial complexes are defined with some orientation, and therefore the value $\mathbf{B}_k(i,j)$ is either $-1$ or $1$ if the $k$-simplex $i$ is incident to the $k-1$-simplex $j$, depending on the orientation, and 0 otherwise. Please notice that $\mathbf{B}_0$ is not defined. We define the $k$-*Hodge Laplacians* as $\mathbf{L}_k = \mathbf{B}_k^\top \mathbf{B}_k + \mathbf{B}_{k+1}\mathbf{B}_{k+1}^\top$, where $\mathbf{L}_{k,d} = \mathbf{B}_k^\top \mathbf{B}_k$ is the *lower Laplacian*, $\mathbf{L}_{k,u} = \mathbf{B}_{k+1}\mathbf{B}_{k+1}^\top$ is the *upper Laplacian*, $\mathbf{L}_0 = \mathbf{B}_1\mathbf{B}_1^\top$ is the *graph Laplacian*. Discrete SNNs [18, 17] define their convolution operations as matrix polynomials of the Hodge Laplacians over *simplicial signals*, *i.e.*, signals defined over the simplicial complex.

**Simplicial signal.** We define a $k$-*simplicial signal* as a function in $\mathcal{X}_k$ as $x_k : \mathcal{X}_k \to \mathbb{R}$. Therefore, we can define a one-dimensional $k$-simplicial signal as $\mathbf{x}_k \in \mathbb{R}^{|\mathcal{X}_k|}$. We can calculate how $\mathbf{x}_k$ varies *w.r.t.* the faces and cofaces of $k$-simplices by $\mathbf{B}_k\mathbf{x}_k$ and $\mathbf{B}_{k+1}^\top\mathbf{x}_k$ [18]. For example, in a node signal $\mathbf{x}_0$, $\mathbf{B}_1^\top\mathbf{x}_0$ computes its *gradient* as the difference between adjacent nodes, and in an edge signal $\mathbf{x}_1$, $\mathbf{B}_1\mathbf{x}_1$ computes its *divergence* [18].

**Dirichlet energy.** The Dirichlet energy $E(\cdot)$ quantifies the smoothness of a simplicial signal with respect to the $k$-Hodge Laplacian [18], where a lower energy value indicates a smoother signal.

**Definition 3.1** (from [18]). The Dirichlet energy of a simplicial signal $\mathbf{x}_k$ can be stated as:

$$E(\mathbf{x}_k) := \mathbf{x}_k^\top \mathbf{L}_k \mathbf{x}_k = \|\mathbf{B}_k\mathbf{x}_k\|_2^2 + \|\mathbf{B}_{k+1}^\top\mathbf{x}_k\|_2^2. \tag{1}$$

This definition generalizes the Dirichlet energy from graphs to simplicial complexes, measuring how similar the values assigned to adjacent simplices are, with higher energy indicating larger variation.

**Simplicial filters.** For a $k$-simplicial signal $\mathbf{x}_k$, a simplicial filter is a function $f : \mathbb{R}^{|\mathcal{X}_k|} \to \mathbb{R}^{|\mathcal{X}_k|}$ as:

$$f(\mathbf{x}_k) = \left( \sum_{i=0}^{T_d} \alpha_i \mathbf{L}_{k,d}^i + \sum_{i=0}^{T_u} \beta_i \mathbf{L}_{k,u}^i \right) \mathbf{x}_k, \tag{2}$$

where $\{\alpha_0, \ldots, \alpha_{T_d}\}$ and $\{\beta_0, \ldots, \beta_{T_u}\}$ are the parameters of the polynomials, and $T_d$, $T_u$ are the order of the polynomials [34]. Please notice that the well-known graph filter [35] is a specific case of (2). In this case, the graph signal is given by $\mathbf{x}_0 \in \mathbb{R}^{|\mathcal{X}_0|}$ and since $\mathbf{B}_0$ is not defined, we have $f(\mathbf{x}_0) = \sum_{i=0}^{T_u} \beta_i \mathbf{L}_0^i \mathbf{x}_i$, *i.e.*, the classical graph filter with the graph Laplacian as the shift operator.

## 3.2 Discrete Simplicial Neural Network

Analogous to a convolutional neural network, an SNN can be defined for simplicial complexes using simplicial filters [17]. However, notice that relying only on filters like in (2) ignores the connections among the adjacent simplices modeled by $\mathbf{L}_k$. Since different simplicial signals influence each other via the simplicial complex localities, previous works have defined simplicial filter banks [34]. The following discrete Hodge-aware filtering model is adapted from SCCNNs [18].

**One-dimensional case.** Let the lower and upper projections of a simplicial signal $\mathbf{x}_k^l$ at layer $l$ be $\mathbf{x}_{k,d}^l = \mathbf{B}_k^\top \mathbf{x}_{k-1}^l \in \mathbb{R}^{|\mathcal{X}_k|}$ and $\mathbf{x}_{k,u}^l = \mathbf{B}_{k+1} \mathbf{x}_{k+1}^l \in \mathbb{R}^{|\mathcal{X}_k|}$, respectively[1]. We define a simplicial layer (with parameters $\theta$ and $\psi$ [18]) as a function $g : \mathbb{R}^{|\mathcal{X}_k|} \times \mathbb{R}^{|\mathcal{X}_k|} \times \mathbb{R}^{|\mathcal{X}_k|} \to \mathbb{R}^{|\mathcal{X}_k|}$ given as:

$$\mathbf{x}_k^l = \sigma \left( \mathbf{H}_{k,d}^l \mathbf{x}_{k,d}^{l-1} + \mathbf{H}_k^l \mathbf{x}_k^{l-1} + \mathbf{H}_{k,u}^l \mathbf{x}_{k,u}^{l-1} \right), \tag{3}$$

where $\mathbf{H}_{k,d}^l := \sum_{i=0}^{T_d} \theta_{k,d,i}^l \mathbf{L}_{k,d}^i$, $\mathbf{H}_{k,u}^l := \sum_{i=0}^{T_u} \theta_{k,u,i}^l \mathbf{L}_{k,u}^i$ and $\mathbf{H}_k^l := \sum_{i=0}^{T_d} \psi_{k,d,i}^l \mathbf{L}_{k,d}^i + \sum_{i=0}^{T_u} \psi_{k,u,i}^l \mathbf{L}_{k,u}^i$.

**Multi-dimensional case.** Let $\{\mathbf{X}_k^l, \mathbf{X}_{k,d}^l, \mathbf{X}_{k,u}^l\}$ be the $F_{l-1}$-dimensional simplicial signal and its lower and upper projections at layer $l$. Let $\boldsymbol{\Theta}_{k,d,i}^l$, $\boldsymbol{\Theta}_{k,u,i}^l$, $\boldsymbol{\Psi}_{k,d,i}^l$, and $\boldsymbol{\Psi}_{k,u,i}^l$ be learnable linear projections in $\mathbb{R}^{F_{l-1} \times F_l}$ corresponding to the $\alpha$ and $\phi$ parameters for the unidimensional case in (3). Using (2) and (3), we can define an SNN layer for the multidimensional case as follows [18]:

$$\mathbf{X}_k^l = \sigma \left( \sum_{i=0}^{T_d} \mathbf{L}_{k,d}^i \mathbf{X}_{k,d}^{l-1} \boldsymbol{\Theta}_{k,d,i}^l + \sum_{i=0}^{T_d} \mathbf{L}_{k,d}^i \mathbf{X}_k^{l-1} \boldsymbol{\Psi}_{k,d,i}^l + \sum_{i=0}^{T_u} \mathbf{L}_{k,u}^i \mathbf{X}_k^{l-1} \boldsymbol{\Psi}_{k,u,i}^l + \sum_{i=0}^{T_u} \mathbf{L}_{k,u}^i \mathbf{X}_{k,u}^{l-1} \boldsymbol{\Theta}_{k,u,i}^l \right). \tag{4}$$

The discrete SNN in (4) is analogous to the GNN case, where discrete powers of the Hodge Laplacians capture multi-hop diffusions in the simplicial signal and its lower and upper projections.

All the proofs of theorems, propositions, and lemmas of the paper are provided in Appendices A-H.

# 4 Continuous Simplicial Neural Network

Discrete SNNs provide flexibility in filtering simplicial signals through lower and upper projections. However, their information propagation remains fixed for each polynomial order, limiting adaptability. In this section, we introduce COSIMO, which enables a dynamic receptive field in each convolutional operation. We begin by formulating the PDEs that govern physics-informed dynamics over simplicial complexes. Next, we define the fundamental operations of COSIMO as the solutions to these PDEs. Finally, we provide a rigorous stability analysis of COSIMO, showing its robustness to topological perturbations in simplicial complexes.

## 4.1 PDEs in Simplicial Complexes

Our set of PDEs is inspired by heat diffusion on simplicial complexes, providing a natural extension of discrete SNNs. Intuitively, performing heat diffusion over the decoupled Hodge Laplacians enables information propagation at different rates within the continuous domain of the simplicial complex. This parallels the case in graphs [19], where continuous GNN formulations have been shown to generalize certain discrete GNNs [36], opening new possibilities for architectural design.

In our framework, considering both joint diffusion on $s^k$ and independent diffusion on $s^{k-1}$ and $s^{k+1}$ allows for greater flexibility in modeling complex relationships. By enabling these dynamics to evolve at different rates, we can better adapt to the underlying topology. Motivated by these considerations, we model simplicial heat diffusion using a system of PDEs on the Hodge Laplacians. Let $t_d$ and $t_u$

---

[1]In this work, the superscript $l$ refers to the layer index and should not be confused with exponentiation.

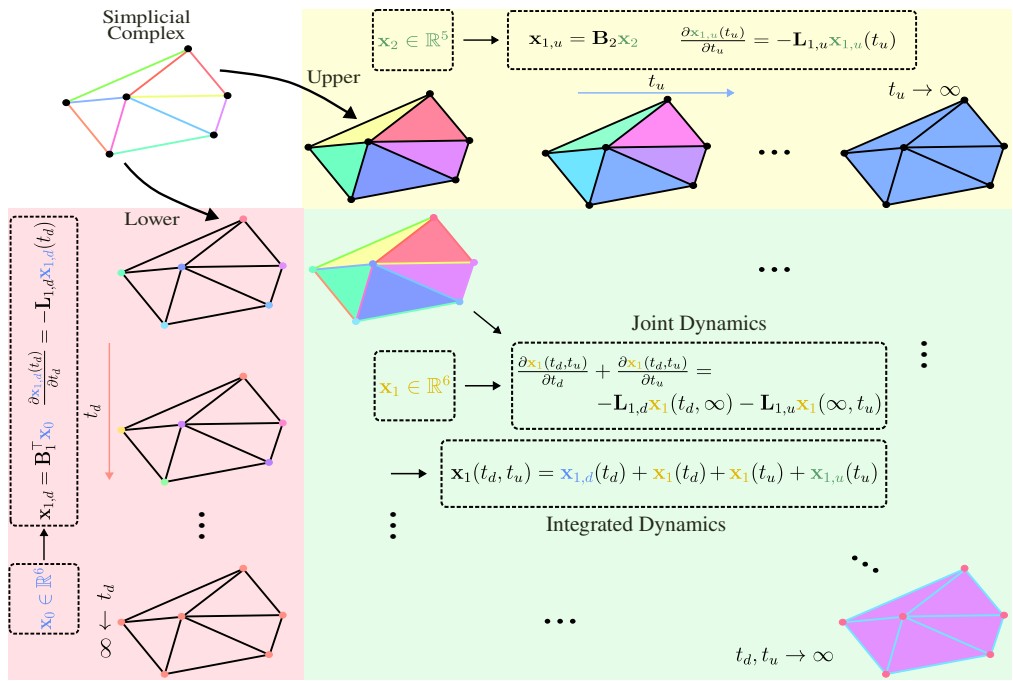

Figure 2: The PDE-based signal evolution on a simplicial complex, governed by independent diffusion processes on the lower and upper Hodge Laplacians and a coupled process integrating both spaces. The colors in the simplicial complexes represent the values of the underlying simplicial signals.

be the time variables governing the dynamics in the lower and upper Laplacians, respectively. We define these dynamics through the following system of PDEs:

*Independent lower dynamics:* The signal evolution in the lower space follows a heat diffusion process:

$$\frac{\partial \mathbf{x}_{k,d}(t_d)}{\partial t_d} = -\mathbf{L}_{k,d}\mathbf{x}_{k,d}(t_d). \tag{5}$$

*Independent upper dynamics:* Similarly, the signal in the upper space evolves according to:

$$\frac{\partial \mathbf{x}_{k,u}(t_u)}{\partial t_u} = -\mathbf{L}_{k,u}\mathbf{x}_{k,u}(t_u). \tag{6}$$

*Joint dynamics:* The interaction between the lower and upper spaces is captured by:

$$\frac{\partial \mathbf{x}_k(t_d, t_u)}{\partial t_d} + \frac{\partial \mathbf{x}_k(t_d, t_u)}{\partial t_u} = -\mathbf{L}_{k,d}\mathbf{x}_k(t_d, \infty) - \mathbf{L}_{k,u}\mathbf{x}_k(\infty, t_u), \tag{7}$$

where $\mathbf{x}_k(t_d, \infty) = \lim_{t_u \to \infty} \mathbf{x}_k(t_d, t_u)$ and $\mathbf{x}_k(\infty, t_u) = \lim_{t_d \to \infty} \mathbf{x}_k(t_d, t_u)$ state marginal stable solutions in upper and lower subspaces.

*Integrated dynamics:* The final solution by integration of the independent and joint dynamics is as:

$$\mathbf{x}_k(t_d, t_u) = \mathbf{x}_{k,d}(t_d) + \mathbf{x}_k(t_d) + \mathbf{x}_k(t_u) + \mathbf{x}_{k,u}(t_u). \tag{8}$$

These dynamics describe the information flow across different simplicial levels, ensuring a principled integration of independent and coupled dynamics. Refer to Fig. 2 for a visual representation on $s^2$.

*Remark* 4.1. With $t_d = t_u = t$, the joint dynamic PDE in (7) turns into $\frac{\partial \mathbf{x}_k(t)}{\partial t} = -\mathbf{L}_{k,d}\mathbf{x}_k(t) - \mathbf{L}_{k,u}\mathbf{x}_k(t)$, and its steady state solution, $\mathbf{L}_k\mathbf{x}_k = \mathbf{0}$, lies in the kernel space of $\mathbf{L}_k$ [18], justifying the need for independent lower and upper Hodge-aware PDEs.

## 4.2 COSIMO as a Solution to the Simplicial PDEs

We propose COSIMO as a solution to the descriptive sets of PDEs introduced in Section 4.1.

**Proposition 4.2.** *The solution to the descriptive sets of PDEs in Section 4.1 is given by:*

$$\mathbf{x}'_k(t_d, t_u) = \overbrace{e^{-t_d \mathbf{L}_{k,d}} \mathbf{x}_{k,d}(0)}^{\mathbf{x}_{k,d}(t_d)} + \overbrace{e^{-t_u \mathbf{L}_{k,u}} \mathbf{x}_{k,u}(0)}^{\mathbf{x}_{k,u}(t_u)} + \overbrace{e^{-t_d \mathbf{L}_{k,d}} \mathbf{x}_k(0,0) + e^{-t_u \mathbf{L}_{k,u}} \mathbf{x}_k(0,0)}^{\mathbf{x}_k(t_d,t_u)}, \quad (9)$$

*where* $\mathbf{x}_{k,d}(0)$, $\mathbf{x}_{k,u}(0)$, *and* $\mathbf{x}_k(0,0)$ *are the initial conditions for the PDEs.*

Using (9) and extending the solution to the multidimensional case, we propose the $l$-th layer of COSIMO as (considering $\sigma(\cdot)$ as a nonlinearity and $\{\mathbf{X}_k^0, \mathbf{X}_{k,d}^0, \mathbf{X}_{k,u}^0\}$ as the initial conditions):

$$\mathbf{X}_k^l = \sigma\Big(e^{-t_d \mathbf{L}_{k,d}} \mathbf{X}_{k,d}^{l-1} \mathbf{\Theta}_{k,d}^l + e^{-t_u \mathbf{L}_{k,u}} \mathbf{X}_{k,u}^{l-1} \mathbf{\Theta}_{k,u}^l + e^{-t_d \mathbf{L}_{k,d}} \mathbf{X}_k^{l-1} \mathbf{\Psi}_{k,d}^l + e^{-t_u \mathbf{L}_{k,u}} \mathbf{X}_k^{l-1} \mathbf{\Psi}_{k,u}^l\Big), \quad (10)$$

where $\{\mathbf{\Theta}_{k,d}^l, \mathbf{\Theta}_{k,u}^l, \mathbf{\Psi}_{k,d}^l, \mathbf{\Psi}_{k,u}^l\}$ are learnable linear projections in $\mathbb{R}^{F_{l-1} \times F_l}$.

*Remark* 4.3. One possible approach to make our model more expressive is the aggregation of $M$ learnable branches. Let $f_{k,l}^{(m)}(\mathbf{X}; t, \mathbf{\Theta}, \mathbf{\Psi})$ be the $m$-th branch of layer $l$ as the right-hand side of the COSIMO model in (10), where $\mathbf{X} = \{\mathbf{X}_k^0, \mathbf{X}_{k,d}^0, \mathbf{X}_{k,u}^0\}$, $t^{(m)} = \{t_d^{(m)}, t_u^{(m)}\}$, $\mathbf{\Theta}^{(m)} = \{\mathbf{\Theta}_{k,d}^{l,(m)}, \mathbf{\Theta}_{k,u}^{l,(m)}\}$, and $\mathbf{\Psi}^{(m)} = \{\mathbf{\Psi}_{k,d}^{l,(m)}, \mathbf{\Psi}_{k,u}^{l,(m)}\}$. Considering `AGG(.)` as a well-defined aggregation function, *e.g.*, a multilayer perceptron, the output for the $l$-th layer can be stated as:

$$\mathbf{X}_k^l = \texttt{AGG}(\{f_{k,l}^{(m)}(\mathbf{X}; t, \mathbf{\Theta}, \mathbf{\Psi})\}_{m=1}^M). \quad (11)$$

### 4.3 Computational Complexity of COSIMO

For the efficient implementation of exponential Hodge filters, we benefit from the eigenvalue decomposition (EVD) of the Hodge Laplacians [37, 21]. Precisely, for a Laplacian $\mathbf{L} \in \mathbb{R}^{N \times N}$ with the eigenvalues $\lambda_0 = 0 \leq \lambda_1 \leq \ldots \leq \lambda_{N-1}$, after performing EVD, $\mathbf{L} = \mathbf{V} \mathbf{\Lambda} \mathbf{V}^\top$, the exponential Laplacian filtering on input $\mathbf{X} \in \mathbb{R}^{N \times F}$ and learnable weight $\mathbf{W} \in \mathbb{R}^{F \times F'}$ can be implemented as:

$$e^{-t\mathbf{L}} \mathbf{X} \mathbf{W} \approx \mathbf{V}^{(K)} (\overbrace{[\tilde{\boldsymbol{\lambda}}^{(K)} | \ldots | \tilde{\boldsymbol{\lambda}}^{(K)}]}^{F \text{ times}} \odot (\mathbf{V}^{(K)^\top} \mathbf{X})) \mathbf{W}, \quad (12)$$

where $\tilde{\boldsymbol{\lambda}} := e^{-t \boldsymbol{\lambda}^{(K)}} = (e^{-t \lambda_{N-1}}, \ldots, e^{-t \lambda_{N-K}})^\top$, and $\mathbf{V}^{(K)} \in \mathbb{R}^{N \times K}$ is built by choosing the $K$ most dominant eigenvalue-eigenvector pairs of $\mathbf{L}$, and $\odot$ states the element-wise multiplication. As $K$ gets closer to $N$, the estimation is more accurate. Using the implementation in (12), the computational complexity of the EVD decreases from $\mathcal{O}(N^3)$ to $\mathcal{O}(KN^2)$. Thus, the computational complexity of calculating the Hodge-aware exponential Hodge filters in (9) can be reduced from $\mathcal{O}(|\mathcal{X}_k|^3)$ to $\mathcal{O}(|\mathcal{X}_k|^2 (K_k^{(d)} + K_k^{(u)} + K_k))$, where $K_k^{(d)}$, $K_k^{(u)}$, and $K_k$ are the most dominant eigenvalue-eigenvectors of $\mathbf{L}_{k,d}$, $\mathbf{L}_{k,u}$, and $\mathbf{L}_k$, respectively. The empirical analysis regarding the trade-off on runtime, computational complexity, and performance is presented in the Appendix I.

### 4.4 Stability Analysis

Here, we study the robustness of our model against simplicial perturbations. We model these perturbations as structural inaccuracies in the incidence matrices, given by the additive error models $\tilde{\mathbf{B}}_k = \mathbf{B}_k + \mathbf{E}_k$ and $\tilde{\mathbf{B}}_{k+1} = \mathbf{B}_{k+1} + \mathbf{E}_{k+1}$, where $\|\mathbf{E}_k\| \leq \epsilon_k$ and $\|\mathbf{E}_{k+1}\| \leq \epsilon_{k+1}$ represent the perturbation errors. Building upon these additive models on the continuous simplicial filtering operations in (9), the following theorem bounds the COSIMO's stability:

**Theorem 4.4.** *Given the additive simplicial perturbation models* $\tilde{\mathbf{B}}_k = \mathbf{B}_k + \mathbf{E}_k$ *and* $\tilde{\mathbf{B}}_{k+1} = \mathbf{B}_{k+1} + \mathbf{E}_{k+1}$, *where* $\|\mathbf{E}_k\| \leq \epsilon_k$ *and* $\|\mathbf{E}_{k+1}\| \leq \epsilon_{k+1}$, *the error between true and perturbed targets in (9), i.e.,* $\delta_{\mathbf{X}_k} := \|\tilde{\mathbf{X}}_k(t_d, t_u) - \mathbf{X}_k(t_d, t_u)\|$, *is bounded as:*

$$\delta_{\mathbf{X}_k} \leq t_d \delta_{k,d} e^{t_d \delta_{k,d}} (\|\mathbf{x}_{k,d}(0)\| + \|\mathbf{x}_k(0,0)\|) + t_u \delta_{k,u} e^{t_u \delta_{k,u}} (\|\mathbf{x}_{k,u}(0)\| + \|\mathbf{x}_k(0,0)\|), \quad (13)$$

*where* $\delta_{k,d} := 2\sqrt{\lambda_{max}(\mathbf{L}_{k,d})} \epsilon_k + \epsilon_k^2$, $\delta_{k,u} := 2\sqrt{\lambda_{max}(\mathbf{L}_{k,u})} \epsilon_{k+1} + \epsilon_{k+1}^2$.

Theorem 4.4 shows how the robustness of the model is influenced by the maximum eigenvalues of $\mathbf{L}_{k,d}$ and $\mathbf{L}_{k,u}$, as well as by the Hodge receptive fields $t_d$ and $t_u$. Furthermore, the error bounds of $\epsilon_k$ and $\epsilon_{k+1}$ play a critical role in determining $\delta_{k,d}$ and $\delta_{k,u}$, which ultimately control the stability. The following corollary simplifies Theorem 4.4 under the assumption of sufficiently small error bounds.

**Corollary 4.5.** *When the error bounds $\epsilon_k$ and $\epsilon_{k+1}$ are sufficiently small, the error between the true and perturbed targets is given by $\delta_{\mathbf{X}_k} = \mathcal{O}(\epsilon_k) + \mathcal{O}(\epsilon_{k+1})$.*

Corollary 4.5 demonstrates stability of the proposed network against small simplicial perturbations, generalizing the stability results on the continuous GNNs [20].

## 5 Understanding Over-smoothing in SNNs

We first comprehensively analyze the over-smoothing problem in discrete SNNs. Next, we study over-smoothing in COSIMO highlighting the key differences with discrete SNNs. In both cases, we focus on the Dirichlet energy convergence to zero, making the simplicial signals non-discriminative. In this section, we set $F_l = F$ for all $l$.

### 5.1 Over-smoothing in Discrete SNNs

Based on the discrete SNN in (4) with $T_d = T_u = 1$ (and zeroing weights for $i = 0$) and Definition 3.1, the following theorem characterizes the over-smoothing properties of the discrete SNN.

**Theorem 5.1.** *In the discrete SNN in (4) and nonlinearity functions $\mathrm{ReLU}(\cdot)$ or $\mathrm{LeakyReLU}(\cdot)$, the Dirichlet energy of the simplicial signals at the $l+1$-th layer is bounded by the Dirichlet energy of the $l$-th layer and some structural and architectural characteristics as:*

$$E(\mathbf{X}_k^{l+1}) \leq$$
$$s\tilde{\lambda}_{max}^2 E(\mathbf{X}_k^l) + s\tilde{\lambda}_{max}^3 \left( E_{k-1}(\mathbf{X}_{k-1}^l) + E_{k+1}(\mathbf{X}_{k+1}^l) \right) + 2Fs\tilde{\lambda}_{max}^{3.5}\|\mathbf{X}_k^l\| \left( \|\mathbf{X}_{k-1}^l\| + \|\mathbf{X}_{k+1}^l\| \right),$$
$$\tag{14}$$

*where* $\tilde{\lambda}_{max} := \max_k \{\lambda_{max}(\mathbf{L}_{k,d}), \lambda_{max}(\mathbf{L}_{k,u})\}$, *and* $s :=$ $\sqrt{\max_{k,l,i} \{\|\mathbf{\Theta}_{k,d,i}^l\|, \|\mathbf{\Theta}_{k,u,i}^l\|, \|\mathbf{\Psi}_{k,d,i}^l\|, \|\mathbf{\Psi}_{k,u,i}^l\|\}}$.

The upper bound in (14) is composed of three terms. Unlike GNN counterparts that depend solely on $E(\mathbf{X}_k^l)$ [38], this bound includes two additional terms, the second and third, which help prevent the upper bound from vanishing exponentially. This robustness has been previously analyzed in the context of SCCNNs [18] (more details in Appendix M.2). However, under some practically justified conditions, the upper bound in (14) can converge to zero, *i.e.*, the over-smoothing phenomenon, as described in the next corollary:

**Corollary 5.2.** *In (14), if $\tilde{\lambda}_{max} < \min \{s^{-\frac{1}{3}}, (2Fs)^{-\frac{1}{3.5}}, s^{-\frac{1}{2}}\}$, then $\lim_{l\to\infty} E(\mathbf{X}_k^l) \to 0$.*

With the assumption in Corollary 5.2, we should modify $\tilde{\lambda}_{max}$ to control the upper bound in (14). This involves modifying the structural properties of the simplicial complex. Therefore, preventing discrete SNNs from converging to the over-smoothing state is not straightforward.

### 5.2 Over-smoothing in COSIMO

The next theorem characterizes the counterpart of Theorem 5.1 in the continuous settings.

**Theorem 5.3.** *Considering the continuous Hodge-aware framework in (10) and nonlinearity functions $\mathrm{ReLU}(\cdot)$ or $\mathrm{LeakyReLU}(\cdot)$, and defining $\varphi := \min_k \{t_d\lambda_{min}(\mathbf{L}_{k,d}), t_u\lambda_{min}(\mathbf{L}_{k,u})\}$, it holds:*

$$E(\mathbf{X}_k^{l+1}) \leq s.(e^{-2\varphi} + 1).E(\mathbf{X}_k^l) + s.e^{-2\varphi}.\tilde{\lambda}_{max}.(E(\mathbf{X}_{k-1}^l) + E(\mathbf{X}_{k+1}^l))$$
$$+ 2F.s.(e^{-\varphi} + e^{-2\varphi}).\tilde{\lambda}_{max}^{1.5}.\|\mathbf{X}_k^l\|.(\|\mathbf{X}_{k-1}^l\| + \|\mathbf{X}_{k+1}^l\|) + 2F.s.e^{-\varphi}.\tilde{\lambda}_{max}.\|\mathbf{X}_k^l\|^2,$$
$$\tag{15}$$

*where $s := \sqrt{\max_{k,l} \{\|\mathbf{\Theta}_{k,d}^l\|, \|\mathbf{\Theta}_{k,u}^l\|, \|\mathbf{\Psi}_{k,d}^l\|, \|\mathbf{\Psi}_{k,u}^l\|\}}$.*

We observe that the first two terms in (15) tend to converge to zero as in (14) when stacking multiple layers. However, the third term might have a different behavior described in the following corollary.

**Corollary 5.4.** *The upper bound in (15) exponentially converges to zero by stacking layers if*
$$\ln(s) < \min \left\{ -\ln(1 + e^{-2\varphi}), 2\varphi - \ln(\tilde{\lambda}_{max}), \varphi - \ln(2F(1 + e^{-\varphi})\tilde{\lambda}_{max}^{1.5}), \varphi - \ln(2F\tilde{\lambda}_{max}) \right\}.$$

Table 1: Accuracies in trajectory prediction on synthetic and `ocean-drifts` datasets.

| Method | synthetic ↑ | ocean-drifts ↑ |
|---|---|---|
| SNN [14] | $0.655 \pm 0.02$ | $0.525 \pm 0.06$ |
| SCoNe [26] | $0.631 \pm 0.03$ | $0.490 \pm 0.08$ |
| SCNN [17] | $\mathbf{0.677 \pm 0.02}$ | $0.530 \pm 0.08$ |
| Bunch [31] | $0.623 \pm 0.04$ | $0.460 \pm 0.06$ |
| SCCNN [18] | $0.652 \pm 0.04$ | $\underline{0.545 \pm 0.08}$ |
| COSIMO | $\underline{0.659 \pm 0.04}$ | $\mathbf{0.550 \pm 0.06}$ |

Table 2: MSE in regression on partial deformable shapes on the `Shrec-16` dataset.

| Method | small ↓ | full ↓ |
|---|---|---|
| HSN [43] | $0.138 \pm 0.001$ | $0.133 \pm 0.001$ |
| SCACMPS [24] | $0.137 \pm 0.011$ | $0.432 \pm 0.001$ |
| SAN [11] | $0.052 \pm 0.011$ | $0.075 \pm 0.002$ |
| SCCNN [18] | $\underline{0.020 \pm 0.003}$ | $\underline{0.063 \pm 0.003}$ |
| COSIMO | $\mathbf{0.010 \pm 0.004}$ | $\mathbf{0.027 \pm 0.007}$ |

Assuming $t_d = t_u = t$ in (9) (then $\varphi = t\lambda_{\min}(\mathbf{L}_k)$) and considering $t$ as a hyperparameter, one heuristic to prevent over-smoothing in COSIMO is stated in the next proposition.

**Proposition 5.5.** *If* $\ln(s) > 2\varphi - \ln(\tilde{\lambda}_{max})$ *(violating one of the conditions in Corollary 5.4), then* $t < \frac{\ln(s\tilde{\lambda}_{max})}{2\lambda_{min}(\mathbf{L}_k)} + k_f(\mathbf{L}_k)$, *where* $k_f(\mathbf{L}_k)$ *is the finite condition number [39] of the $k$-simplex.*

Theorem 5.3 aligns with the main takeaways in the GNN literature [40, 21], where increasing the graph receptive field leads to an increase in the mixing rate of the node features, leading to a faster convergence to the over-smoothing state. We observe from Proposition 5.5 that decreasing the simplicial receptive field $t$ can alleviate over-smoothing, which is a key difference from the discrete case discussed in Section 5.1. We experimentally validate this claim in Section 6. Besides stability and over-smoothing, we show the permutation equivariance property of COSIMO in Appendix J.

*Remark* 5.6. Due to the differentiability of our model in (10), the simplicial receptive fields $\{t_d, t_u\}$ can be treated as learnable parameters [37], which is the case with all of our experiments except Section 6.2. This provides a significant advantage over the discrete SNN formulation in (3), where the graph filter orders $\{T_d, T_u\}$ must be manually tuned, leading to additional computational cost.

## 6 Experiments and Results

We evaluate COSIMO against SOTA methods in applications of trajectory prediction, mesh regression, node and graph classification. We compare COSIMO with a wide range of graph and simplicial models, including GCN [3], GraphSAGE [41], GIN [32], GAT [42], SNN [14], SCoNe [26], SCNN [17], Bunch [31], HSN [43], SCACMPS [24], SAN [11], SaNN [44], GSAN [45] and SCCNN [18]. Then, we experimentally validate the theoretical claims in this paper. More details on the datasets and implementation are outlined in the Appendix K and L, respectively.

### 6.1 Real-world Applications

**Trajectory prediction.** Trajectory prediction involves forecasting paths within simplicial complexes. To evaluate the effectiveness of COSIMO, we assess its performance on two datasets: a synthetic simplicial complex and the `ocean-drifts` dataset from [26, 46]. As shown in Table 1, COSIMO, SCCNN, and Bunch, which incorporate inter-simplicial couplings, do not outperform SCNN on the synthetic dataset. This is likely because the input data assigned to nodes and triangles is zero, as noted in [26], making inter-simplicial couplings ineffective. However, in the `ocean-drifts` dataset, where higher-order information plays a more significant role, incorporating higher-order convolutions—as in COSIMO and SCCNN—improves the average accuracy.

**Regression on partial deformable shapes.** The `Shrec-16` benchmark [47] extends prior mesh classification datasets to meshes with missing parts, where each class has a full template in a neutral pose for evaluation. To increase complexity, all shapes were sampled to 10K vertices before introducing missing parts in two regular and irregular ways. This results in a dataset of 599 shapes across eight classes (humans and animals). The dataset is divided into a training set (199 shapes) and a test set (400 shapes). The main task here is to regress the correct mesh class under missing parts. We compare COSIMO against SOTA methods on two *small* and *full* versions of the dataset. As shown in Table 2, COSIMO achieves the lowest mean square error (MSE) in mesh regression on both dataset versions, outperforming all baselines. Notably, its superior performance on both small and full versions highlights its adaptability to different amounts of data, *e.g.*, even limited data.

Table 3: Accuracy results on node classification (NC) and graph classification (GC) tasks.

| Method | high-school ↑ (NC) | senate-bills ↑ (NC) | proteins ↑ (GC) |
|---|---|---|---|
| GCN [3] | $0.40 \pm 0.04$ | $0.67 \pm 0.06$ | $0.58 \pm 0.05$ |
| GraphSAGE [41] | $0.27 \pm 0.05$ | $0.54 \pm 0.03$ | $0.61 \pm 0.03$ |
| GIN [32] | $0.18 \pm 0.04$ | $0.53 \pm 0.04$ | $0.61 \pm 0.03$ |
| GAT [42] | $0.34 \pm 0.05$ | $0.50 \pm 0.04$ | $0.57 \pm 0.06$ |
| SCNN [17] | $0.81 \pm 0.01$ | $0.62 \pm 0.05$ | $0.61 \pm 0.03$ |
| SCCNN [18] | $0.88 \pm 0.04$ | $0.64 \pm 0.09$ | $0.69 \pm 0.06$ |
| SAN [29, 11] | $0.86 \pm 0.04$ | $0.53 \pm 0.09$ | $0.64 \pm 0.05$ |
| SaNN [44] | $0.83 \pm 0.03$ | $0.61 \pm 0.08$ | $0.77 \pm 0.02$ |
| GSAN [45] | $0.88 \pm 0.05$ | *OOM* | $0.77 \pm 0.04$ |
| COSIMO | $\mathbf{0.90 \pm 0.05}$ | $\mathbf{0.69 \pm 0.08}$ | $\mathbf{0.79 \pm 0.01}$ |

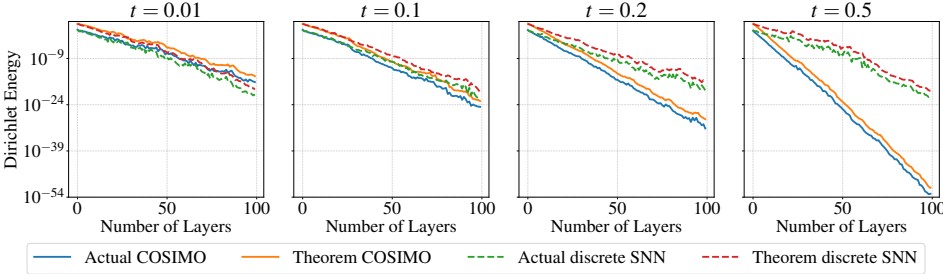

Figure 3: Over-smoothing results of discrete SNNs and COSIMO across different layer depths.

**Node classification (NC).** In this task, the objective is to infer categorical labels associated with 0-dimensional simplices (nodes) embedded in a simplicial complex on two benchmark datasets: `high-school` [48, 49], and `senate-bills` [48, 50, 51]. We partition each dataset chronologically, reserving the initial 80% for training the encoder and the final 20% for evaluation, as in the literature [44, 52]. Table 3 shows the results for node classification. We observe that COSIMO outperforms previous simplicial and graph-based SOTA methods.

**Graph classification (GC).** This task entails assigning discrete labels to entire graphs that result from a clique-lifting process applied to simplicial complexes conducted on the standard `proteins` dataset [53]. Model performance is assessed by computing the average classification accuracy over ten stratified folds [44, 52, 45] in Table 3. Similar to the NC task, COSIMO obtains superior results by leveraging the higher-order information. In fact, the Hodge-aware SNNs are mostly effective in cases of existing higher-order information (*e.g.*, on the edges, triangles, etc.). But, the poor performance of GNNs is probably due to relying only on the data over 0-order simplices, *i.e.*, the nodes, and neglecting the other simplices like edge flows or triangular dynamics.

## 6.2 Over-smoothing Analysis

The goals of this section are twofold: (i) to validate Theorems 5.1 and 5.3, and (ii) to study the behavior of discrete SNNs in (4) and COSIMO in (10) when facing over-smoothing. For the discrete SNN, we consider $T_d = T_u = 1$ ($i = 1$) in (4). For COSIMO in (10), we explore different scenarios by setting the receptive fields $t_d = t_u = t$ where $t \in \{10^{-2}, 10^{-1}, 0.2, 0.5\}$. In both cases, the linear projections with hidden units $F_{l-1} = F_l = 4$ are generated from normal distributions. Figure 3 shows the left-hand side (LHS) and right-hand side (RHS) of Theorems 5.1 and 5.3 averaged over 50 random realizations with number of layers varying from 1 to 100. These results validate Theorems 5.1 and 5.3, confirming that the LHSs are upper bounded by the RHSs. We also observe that adjusting $t$ in COSIMO provides control over the over-smoothing rate, *i.e.*, how quickly the output of the SNN converges to zero Dirichlet energy. Specifically, setting $t = 10^{-2}$ results in a slower over-smoothing rate in COSIMO compared to the discrete SNN. In contrast, increasing $t$ leads to a faster over-smoothing rate in COSIMO than in the discrete SNN. This shows that variations in the continuous receptive fields in (10) directly influence the rate of convergence to the over-smoothing state. Additional analysis has been provided in the Appendix M.

### 6.3 Stability Analysis

We generate 2-order simplicial complexes, following the approach in [26]: $i$) we uniformly sample $N = 30$ random points from the unit square and construct the Delaunay triangulation, $ii$) we remove triangles contained within predefined disk regions, and $iii$) we use the generative model in (9) with $k = 1$, $t_d = 1$, and $t_u = 2$, generating $\{\mathbf{X}_k^0 \in \mathbb{R}^{|\mathcal{X}_k| \times 1}\}_{k=0}^2$ from normal probability distributions. After extracting the incidence matrices $\mathbf{B}_1$ and $\mathbf{B}_2$, we include noise by varying their respective signal-to-noise ratios (SNRs) in $\{-5, 0, 10, 20\}$ dB. For each setting, we train COSIMO and evaluate its prediction performance, averaged over 30 random realizations. Figure 4 presents the results, including standard deviations. The

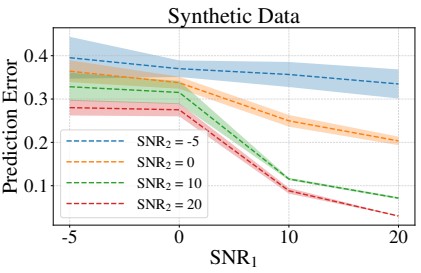

Figure 4: Stability analysis under varying SNRs.

results confirm that the model's overall stability arises from the stability on upper and lower subspaces, validating theoretical findings of Theorem 4.4.

## 7 Conclusion and Limitations

In this paper, we have introduced COSIMO, a novel Hodge-aware model for filtering simplicial signals that addresses the limitations of discrete SNNs by incorporating dynamic receptive fields. We provided rigorous theoretical analyses of the stability and over-smoothing behavior of our model, offering new insights into its performance. Through extensive experiments, we validated our theoretical findings, demonstrating that COSIMO is not only stable but also allows for effective control over the over-smoothing rate through its continuous receptive fields. Our experimental results highlight the superiority of COSIMO over existing SOTA SNNs, particularly in challenging trajectory prediction, regression of partial shapes, node and graph classification tasks.

Regarding future work, although we discussed in detail how to reduce the computational complexity of EVD operations, we will seek to alleviate the need for performing EVDs and potential alternatives to ease this requirement, like non-negative matrix factorization and Cholesky decomposition. Precisely, we will explore the direct implicit Euler method [54] for matrix exponential approximations to reduce the EVD computational complexity at the expense of higher runtime.

## Acknowledgments

This research was supported by DATAIA Convergence Institute as part of the «Programme d'Investissement d'Avenir», (ANR-17-CONV-0003) operated by the center Hi! PARIS. This work was also partially supported by the EuroTech Universities Alliance, and the ANR French National Research Agency under the JCJC projects DeSNAP (ANR-24-CE23-1895-01) and GraphIA (ANR-20-CE23-0009-01).

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

# A  Proof of Proposition 4.2

*Proof.* By considering the proposed solution in (9) as follows:

$$\mathbf{x}'_k(t_d, t_u) = \overbrace{e^{-t_d \mathbf{L}_{k,d}} \mathbf{x}_{k,d}(0)}^{\mathbf{x}_{k,d}(t_d)} + \overbrace{e^{-t_u \mathbf{L}_{k,u}} \mathbf{x}_{k,u}(0)}^{\mathbf{x}_{k,u}(t_u)} + \overbrace{e^{-t_d \mathbf{L}_{k,d}} \mathbf{x}_k(0,0) + e^{-t_u \mathbf{L}_{k,u}} \mathbf{x}_k(0,0)}^{\mathbf{x}_k(t_d, t_u)}, \tag{16}$$

the simplified generative PDE of this formulation can be expressed as:

$$
\begin{aligned}
\frac{\partial \mathbf{x}_{k,d}(t_d)}{\partial t_d} &= -\mathbf{L}_{k,d} e^{-t_d \mathbf{L}_{k,d}} \mathbf{x}_{k,d}(0) = -\mathbf{L}_{k,d} \mathbf{x}_{k,d}(t_d), \\
\frac{\partial \mathbf{x}_{k,u}(t_u)}{\partial t_u} &= -\mathbf{L}_{k,u} e^{-t_u \mathbf{L}_{k,u}} \mathbf{x}_{k,u}(0) = -\mathbf{L}_{k,u} \mathbf{x}_{k,u}(t_u), \\
\mathbf{x}_k(t_d, t_u) &= e^{-t_d \mathbf{L}_{k,d}} \mathbf{x}_k(0,0) + e^{-t_u \mathbf{L}_{k,u}} \mathbf{x}_k(0,0), \\
&\to \lim_{t_d \to \infty} \mathbf{x}_k(t_d, t_u) = e^{-t_u \mathbf{L}_{k,u}} \mathbf{x}_k(0,0), \quad \lim_{t_u \to \infty} \mathbf{x}_k(t_d, t_u) = e^{-t_d \mathbf{L}_{k,d}} \mathbf{x}_k(0,0), \\
&\to \frac{\partial \mathbf{x}_k(t_d, t_u)}{\partial t_d} = -\mathbf{L}_{k,d} e^{-t_d \mathbf{L}_{k,d}} \mathbf{x}_k(0,0), \quad \frac{\partial \mathbf{x}_k(t_d, t_u)}{\partial t_u} = -\mathbf{L}_{k,u} e^{-t_u \mathbf{L}_{k,u}} \mathbf{x}_k(0,0).
\end{aligned}
\tag{17}
$$

Therefore,

$$\frac{\partial \mathbf{x}_k(t_d, t_u)}{\partial t_d} + \frac{\partial \mathbf{x}_k(t_d, t_u)}{\partial t_u} = -\mathbf{L}_{k,d} \overbrace{\mathbf{x}_k(t_d, \infty)}^{\lim_{t_u \to \infty} \mathbf{x}_k(t_d, t_u)} -\mathbf{L}_{k,u} \overbrace{\mathbf{x}_k(\infty, t_u)}^{\lim_{t_d \to \infty} \mathbf{x}_k(t_d, t_u)}, \tag{18}$$

which concludes the proof. $\qquad\qquad\square$

# B  Proof of Theorem 4.4

*Proof.* Using the additive perturbation models and the triangular inequality principle, one can write:

$$
\begin{aligned}
\|\tilde{\mathbf{L}}_{k,d} - \mathbf{L}_{k,d}\| &\le 2\|\mathbf{B}_k\| \, \|\mathbf{E}_k\| + \|\mathbf{E}_k^\top \mathbf{E}_k\| \le 2\epsilon_k \sqrt{\lambda_{\max}(\mathbf{L}_{k,d})} + \epsilon_k^2, \\
\|\tilde{\mathbf{L}}_{k,u} - \mathbf{L}_{k,u}\| &\le 2\|\mathbf{B}_{k+1}\| \, \|\mathbf{E}_{k+1}\| + \|\mathbf{E}_{k+1}^\top \mathbf{E}_{k+1}\| \le 2\epsilon_{k+1} \sqrt{\lambda_{\max}(\mathbf{L}_{k,u})} + \epsilon_{k+1}^2.
\end{aligned}
\tag{19}
$$

Now, for a Laplacian perturbation model on $\mathbf{L}$, one can write [55, 20]

$$\|e^{-t\tilde{\mathbf{L}}} - e^{-t\mathbf{L}}\| \le t\|\tilde{\mathbf{L}} - \mathbf{L}\| \, \|e^{-t\mathbf{L}}\| e^{\|t(\tilde{\mathbf{L}} - \mathbf{L})\|}, \tag{20}$$

for a positive constant $\rho$. Then, by the definitions $\delta_{k,d} := 2\epsilon_k \sqrt{\lambda_{\max}(\mathbf{L}_{k,d})} + \epsilon_k^2$ and $\delta_{k,u} := 2\epsilon_{k+1} \sqrt{\lambda_{\max}(\mathbf{L}_{k,u})} + \epsilon_{k+1}^2$, the bound on the exponential Hodge filters can be obtained as:

$$\|e^{-t_d \tilde{\mathbf{L}}_{k,d}} - e^{-t_d \mathbf{L}_{k,d}}\| \le t_d \overbrace{\|\tilde{\mathbf{L}}_{k,d} - \mathbf{L}_{k,d}\|}^{\delta_{k,d}} \overbrace{\|e^{-t_d \mathbf{L}_{k,d}}\|}^{e^{-t_d(0)}=1} e^{\|t_d(\tilde{\mathbf{L}}_{k,d} - \mathbf{L}_{k,d})\|} \le t_d \delta_{k,d} e^{t_d \delta_{k,d}},$$

$$\|e^{-t_u \tilde{\mathbf{L}}_{k,u}} - e^{-t_u \mathbf{L}_{k,u}}\| \le t_u \overbrace{\|\tilde{\mathbf{L}}_{k,u} - \mathbf{L}_{k,u}\|}^{\delta_{k,u}} \overbrace{\|e^{-t_u \mathbf{L}_{k,u}}\|}^{e^{-t_u(0)}=1} e^{\|t_u(\tilde{\mathbf{L}}_{k,u} - \mathbf{L}_{k,u})\|} \le t_u \delta_{k,u} e^{t_u \delta_{k,u}}. \tag{21}$$

Next, by considering the solution in (9), the perturbation bound can be expressed as:

$$
\begin{aligned}
&\|\tilde{\mathbf{x}}_k(t_d, t_u) - \mathbf{x}_k(t_d, t_u)\| \\
&\le \|e^{-t_d \tilde{\mathbf{L}}_{k,d}} - e^{-t_d \mathbf{L}_{k,d}}\| \, \|\mathbf{x}_{k,d}(0)\| + \|e^{-t_u \tilde{\mathbf{L}}_{k,u}} - e^{-t_u \mathbf{L}_{k,u}}\| \, \|\mathbf{x}_{k,u}(0)\| \\
&+ \left( \|e^{-t_d \tilde{\mathbf{L}}_{k,d}} - e^{-t_d \mathbf{L}_{k,d}}\| + \|e^{-t_u \tilde{\mathbf{L}}_{k,u}} - e^{-t_u \mathbf{L}_{k,u}}\| \right) \|\mathbf{x}_k(0,0)\| \\
&\le t_d \delta_{k,d} e^{t_d \delta_{k,d}} (\|\mathbf{x}_{k,d}(0)\| + \|\mathbf{x}_k(0,0)\|) + t_u \delta_{k,u} e^{t_u \delta_{k,u}} (\|\mathbf{x}_{k,u}(0)\| + \|\mathbf{x}_k(0,0)\|).
\end{aligned}
\tag{22}
$$

$$\square$$

## C    Proof of Corollary 4.5

*Proof.* Based on the proof stated in Appendix B, we can bound $\|\tilde{\mathbf{L}}_k - \mathbf{L}_k\|$ as:

$$
\begin{aligned}
\|\tilde{\mathbf{L}}_k - \mathbf{L}_k\| &\leq \|\tilde{\mathbf{L}}_{k,d} - \mathbf{L}_{k,d}\| + \|\tilde{\mathbf{L}}_{k,u} - \mathbf{L}_{k,u}\| \\
&\leq 2\|\mathbf{B}_k\|\,\|\mathbf{E}_k\| + \|\mathbf{E}_k^\top \mathbf{E}_k\| + 2\|\mathbf{B}_{k+1}\|\,\|\mathbf{E}_{k+1}\| + \|\mathbf{E}_{k+1}\mathbf{E}_{k+1}^\top\| \\
&\leq 2\epsilon_k \sqrt{\lambda_{\max}(\mathbf{L}_{k,d})} + \epsilon_k^2 + 2\epsilon_{k+1}\sqrt{\lambda_{\max}(\mathbf{L}_{k,u})} + \epsilon_{k+1}^2 \\
&\overset{\text{if } \epsilon_k \text{ and } \epsilon_{k+1} \text{ are small}}{\approx} \mathcal{O}(\epsilon_k) + \mathcal{O}(\epsilon_{k+1}).
\end{aligned}
\tag{23}
$$

Using this direction, it can be easily seen that:

$$
\begin{aligned}
\|\tilde{\mathbf{L}}_{k,d} - \mathbf{L}_{k,d}\| &\leq 2\|\mathbf{B}_k\|\,\|\mathbf{E}_k\| + \|\mathbf{E}_k^\top \mathbf{E}_k\| \leq 2\epsilon_k \sqrt{\lambda_{\max}(\mathbf{L}_{k,d})} + \epsilon_k^2 \overset{\text{if } \epsilon_k \text{ is small}}{\approx} \mathcal{O}(\epsilon_k) \\
\|\tilde{\mathbf{L}}_{k,u} - \mathbf{L}_{k,u}\| &\leq 2\|\mathbf{B}_{k+1}\|\,\|\mathbf{E}_{k+1}\| + \|\mathbf{E}_{k+1}^\top \mathbf{E}_{k+1}\| \\
&\leq 2\epsilon_{k+1}\sqrt{\lambda_{\max}(\mathbf{L}_{k,u})} + \epsilon_{k+1}^2 \overset{\text{if } \epsilon_{k+1} \text{ is small}}{\approx} \mathcal{O}(\epsilon_{k+1}).
\end{aligned}
\tag{24}
$$

Now, for a sample Laplacian $\mathbf{L}$ with $\|\tilde{\mathbf{L}} - \mathbf{L}\| = \mathcal{O}(\epsilon)$, one can write:

$$
\|e^{-t\tilde{\mathbf{L}}} - e^{-t\mathbf{L}}\| \leq t\|\tilde{\mathbf{L}} - \mathbf{L}\|\,\|e^{-t\mathbf{L}}\|e^{\|t(\tilde{\mathbf{L}}-\mathbf{L})\|} = \mathcal{O}(\epsilon t e^{-\rho t}) = \mathcal{O}(\epsilon),
\tag{25}
$$

for a positive constant $\rho$ [55, 20]. Therefore, by adapting (25), on can write:

$$
\begin{aligned}
\|e^{-t_d \tilde{\mathbf{L}}_{k,d}} - e^{-t_d \mathbf{L}_{k,d}}\| &\leq t_d\|\tilde{\mathbf{L}}_{k,d} - \mathbf{L}_{k,d}\|\,\|e^{-t_d \mathbf{L}_{k,d}}\|e^{\|t_d(\tilde{\mathbf{L}}_{k,d}-\mathbf{L}_{k,d})\|} \\
&= \mathcal{O}(\epsilon_k t_d e^{-\rho_d t_d}) = \mathcal{O}(\epsilon_k), \\
\|e^{-t_u \tilde{\mathbf{L}}_{k,u}} - e^{-t_u \mathbf{L}_{k,u}}\| &\leq t_u\|\tilde{\mathbf{L}}_{k,u} - \mathbf{L}_{k,u}\|\,\|e^{-t_u \mathbf{L}_{k,u}}\|e^{\|t_u(\tilde{\mathbf{L}}_{k,u}-\mathbf{L}_{k,u})\|} \\
&= \mathcal{O}(\epsilon_{k+1} t_u e^{-\rho_u t_u}) = \mathcal{O}(\epsilon_{k+1}).
\end{aligned}
\tag{26}
$$

By considering (22) and (26), the proof is completed. □

## D    Proof of Theorem 5.1

*Proof.* First, by stating the Dirichlet with $E(.)$, we need the following lemmas from [38]:

**Lemma D.1.** *(Lemma 3.2 in [38]).* $E(\mathbf{XW}) \leq \|\mathbf{W}^\top\|_2^2 E(\mathbf{X})$.

**Lemma D.2.** *(Lemma 3.3 in [38]). For ReLU and Leaky-ReLU nonlinearities* $E(\sigma(\mathbf{X})) \leq E(\mathbf{X})$.

**Lemma D.3.** *(Von Neumann's trace inequality [56]) For two square matrices* $\mathbf{A}$ *and* $\mathbf{B}$ *of size* $m$ *and singular values* $\sigma_i(\mathbf{A})$ *and* $\sigma_i(\mathbf{B})$, *respectively,* $tr(\mathbf{AB}) \leq \sum_{i=1}^m \sigma_i(\mathbf{A})\sigma_i(\mathbf{B}) \leq m\|\mathbf{A}\|_2\|\mathbf{B}\|_2$.

**Lemma D.4.** *Since* $\mathbf{B}_k\mathbf{B}_{k+1} = \mathbf{0}$, *one can obtain* $\mathbf{L}_{k,d}\mathbf{L}_{k,u} = \mathbf{0}$ *and* $\mathbf{L}_{k,d}e^{-t\mathbf{L}_{k,u}} = \mathbf{L}_{k,d}$. *Similar deductions can be obtained for* $\mathbf{L}_{k,u}$.

**Lemma D.5.** *Since* $\mathbf{X}_{k,d} = \mathbf{B}_k^\top \mathbf{X}_{k-1}$ *and* $\mathbf{L}_{k,d} = \mathbf{B}_k^\top \mathbf{B}_k$ *and* $\mathbf{L}_{k-1,u} = \mathbf{B}_k\mathbf{B}_k^\top$, *one can obtain* $E_d(\mathbf{X}_{k,d}) = tr(\mathbf{X}_{k-1}^\top \mathbf{B}_k(\mathbf{B}_k^\top \mathbf{B}_k)\mathbf{B}_k^\top \mathbf{X}_{k-1}) \leq \lambda_{max}(\mathbf{L}_{k-1,u})E_u(\mathbf{X}_{k-1})$. *Similar deductions can be obtained for* $\mathbf{L}_{k,u}$.

Then,

$$
\begin{aligned}
E(\mathbf{X}_k^{l+1}) &= \operatorname{tr}(\mathbf{X}_k^{l+1\top}\mathbf{L}_{k,d}\mathbf{X}_k^{l+1}) + \operatorname{tr}(\mathbf{X}_k^{l+1\top}\mathbf{L}_{k,u}\mathbf{X}_k^{l+1}) \\
&\leq s\lambda_{\max}^2(\mathbf{L}_{k,d})E(\mathbf{X}_{k,d}^l) + 2Fs\lambda_{\max}^3(\mathbf{L}_{k,d})\|\mathbf{X}_{k,d}^l\|_2.\|\mathbf{X}_k^l\|_2 + s\lambda_{\max}^2(\mathbf{L}_{k,d})E_d(\mathbf{X}_k^l) \\
&+ s\lambda_{\max}^2(\mathbf{L}_{k,u})E(\mathbf{X}_{k,u}^l) + 2Fs\lambda_{\max}^3(\mathbf{L}_{k,u})\|\mathbf{X}_{k,u}^l\|_2.\|\mathbf{X}_k^l\|_2 + s\lambda_{\max}^2(\mathbf{L}_{k,u})E_u(\mathbf{X}_k^l) \\
&\leq s\lambda_{\max}^2(\mathbf{L}_{k,d})\lambda_{\max}(\mathbf{L}_{k-1,u})E_u(\mathbf{X}_{k-1}^l) + 2Fs\lambda_{\max}^{3.5}(\mathbf{L}_{k,d})\|\mathbf{X}_k^l\|.\|\mathbf{X}_{k-1}^l\|_2 + s\lambda_{\max}^2(\mathbf{L}_{k,d})E_d(\mathbf{X}_k^l) \\
&+ s\lambda_{\max}^2(\mathbf{L}_{k,u})\lambda_{\max}(\mathbf{L}_{k+1,d})E_d(\mathbf{X}_{k+1}^l) + 2Fs\lambda_{\max}^{3.5}(\mathbf{L}_{k,u})\|\mathbf{X}_k^l\|.\|\mathbf{X}_{k+1}^l\|_2 + s\lambda_{\max}^2(\mathbf{L}_{k,u})E_u(\mathbf{X}_k^l) \\
&\leq s\tilde{\lambda}_{\max}^3 E_u(\mathbf{X}_{k-1}^l) + 2Fs\tilde{\lambda}_{\max}^{3.5}\|\mathbf{X}_k^l\|.\|\mathbf{X}_{k-1}^l\|_2 + s\tilde{\lambda}_{\max}^2 E_d(\mathbf{X}_k^l) \\
&+ s\tilde{\lambda}_{\max}^3 E_d(\mathbf{X}_{k+1}^l) + 2Fs\tilde{\lambda}_{\max}^{3.5}\|\mathbf{X}_k^l\|.\|\mathbf{X}_{k+1}^l\|_2 + s\tilde{\lambda}_{\max}^2 E_u(\mathbf{X}_k^l) \\
&\leq s\tilde{\lambda}_{\max}^3(E(\mathbf{X}_{k-1}^l) + E(\mathbf{X}_{k+1}^l)) + 2Fs\tilde{\lambda}_{\max}^{3.5}\|\mathbf{X}_k^l\|.(\|\mathbf{X}_{k-1}^l\|_2 + \|\mathbf{X}_{k+1}^l\|_2) + s\tilde{\lambda}_{\max}^2 E(\mathbf{X}_k^l).
\end{aligned}
\tag{27}
$$

$\square$

# E  Proof of Corollary 5.2

*Proof.* Using the results of Theorem 5.1, if the constraints of $s\tilde{\lambda}_{\max}^3 < 1$, $2Fs\tilde{\lambda}_{\max}^{3.5} < 1$, and $s\tilde{\lambda}_{\max}^2 < 1$ are simultaneously satisfied, by stacking layers, their multiplications converge to zero making the RHS in Theorem 5.1 converge to zero as well. Holding the mentioned conditions together completes the proof. $\square$

# F  Proof of Theorem 5.3

*Proof.* First, consider that $E(\mathbf{x})$ can be stated by $\tilde{x}$, *i.e.*, the Graph Fourier Transform (GFT) [35] of $\mathbf{x}$ (where $\{\lambda_i\}_{i=1}^N$ eigenvalues of the Laplacian $\hat{\mathbf{L}}$), as follows [38]:

$$
E(\mathbf{x}) = \mathbf{x}^\top \hat{\mathbf{L}} \mathbf{x} = \sum_{i=1}^N \lambda_i \tilde{x}_i^2.
\tag{28}
$$

Next, taking $\lambda$ as the smallest nonzero eigenvalue of the Laplacian $\hat{\mathbf{L}}$, the following lemma describes the behavior of a heat kernel in the most basic scenario of over-smoothing.

**Lemma F.1.** *We have:*

$$
E(e^{-\hat{\mathbf{L}}}\mathbf{x}) \leq e^{-2\lambda}E(\mathbf{x}).
\tag{29}
$$

*Proof.* By showing the EVD of $\hat{\mathbf{L}} = \mathbf{V}\mathbf{\Lambda}\mathbf{V}^\top$ and $e^{-\hat{\mathbf{L}}} = \mathbf{V}e^{-\mathbf{\Lambda}}\mathbf{V}^\top$, we have:

$$
\begin{aligned}
&E(e^{-\hat{\mathbf{L}}}\mathbf{x}) \\
&= \mathbf{x}^\top \overbrace{e^{-\hat{\mathbf{L}}^\top}}^{\mathbf{V}e^{-\mathbf{\Lambda}}\mathbf{V}^\top} \overbrace{\hat{\mathbf{L}}}^{\mathbf{V}\mathbf{\Lambda}\mathbf{V}^\top} \overbrace{e^{-\hat{\mathbf{L}}}}^{\mathbf{V}e^{-\mathbf{\Lambda}}\mathbf{V}^\top} \mathbf{x} = \sum_{i=1}^N \lambda_i \tilde{x}_i^2 e^{-2\lambda_i} \leq e^{-2\lambda}\left(\sum_{i=1}^N \lambda_i \tilde{x}_i^2\right) = e^{-2\lambda}E(\mathbf{x}).
\end{aligned}
\tag{30}
$$

Note that we excluded the zero eigenvalues because they do not engage in the calculation of the Dirichlet energy. $\square$

Leveraging from Lemmas D.1- D.5, and F.1, and also the triangle principle in inequalities, one can simply derive the following useful inequalities:

$$
\boxed{1} \quad E_d(e^{-t_d\mathbf{L}_{k,d}}\mathbf{X}_{k,d}^l\mathbf{\Theta}_{k,d}^l) \leq E_d(e^{-t_d\mathbf{L}_{k,d}}\mathbf{X}_{k,d}^l)\|\mathbf{\Theta}_{k,d}^l\|_2^2
$$

$$
\boxed{2} \quad E_d(e^{-t_d\mathbf{L}_{k,d}}\mathbf{X}_k^l\mathbf{\Psi}_{k,d}^l) \leq E_d(e^{-t_d\mathbf{L}_{k,d}}\mathbf{X}_k^l)\|\mathbf{\Psi}_{k,d}^l\|_2^2
$$

$$
\boxed{3} \quad E_d(e^{-t_d\mathbf{L}_{k,u}}\mathbf{X}_k^l\mathbf{\Psi}_{k,u}^l) \leq E_d(\mathbf{X}_k^l)\|\mathbf{\Psi}_{k,u}^l\|_2^2
$$

$$
\boxed{4} \quad \mathrm{tr}(\mathbf{X}_{k,d}^l{}^\top e^{-t_d\mathbf{L}_{k,d}}\mathbf{L}_{k,d}e^{-t_d\mathbf{L}_{k,d}}\mathbf{X}_k^l\mathbf{\Psi}_{k,d}^l\mathbf{\Theta}_{k,d}^l{}^\top)
$$
$$
\leq F.\|\mathbf{X}_{k,d}^l{}^\top e^{-t_d\mathbf{L}_{k,d}}\mathbf{L}_{k,d}e^{-t_d\mathbf{L}_{k,d}}\mathbf{X}_k^l\|_2\ \|\mathbf{\Psi}_{k,d}^l\mathbf{\Theta}_{k,d}^l{}^\top\|_2
$$

$$
\boxed{5} \quad \mathrm{tr}(\mathbf{X}_{k,d}^l{}^\top e^{-t_d\mathbf{L}_{k,d}}\mathbf{L}_{k,d}e^{-t_u\mathbf{L}_{k,u}}\mathbf{X}_k^l\mathbf{\Psi}_{k,u}^l\mathbf{\Theta}_{k,d}^l{}^\top)
$$
$$
\leq F.\|\mathbf{X}_{k,d}^l{}^\top e^{-t_d\mathbf{L}_{k,d}}\mathbf{L}_{k,d}e^{-t_u\mathbf{L}_{k,u}}\mathbf{X}_k^l\|_2\ \|\mathbf{\Psi}_{k,u}^l\mathbf{\Theta}_{k,d}^l{}^\top\|_2
$$

$$
\boxed{6} \quad \mathrm{tr}(\mathbf{X}_k^l{}^\top e^{-t_d\mathbf{L}_{k,d}}\mathbf{L}_{k,d}e^{-t_u\mathbf{L}_{k,u}}\mathbf{X}_k^l\mathbf{\Psi}_{k,u}^l\mathbf{\Theta}_{k,d}^l{}^\top)
$$
$$
\leq F.\|\mathbf{X}_k^l{}^\top e^{-t_d\mathbf{L}_{k,d}}\mathbf{L}_{k,d}e^{-t_u\mathbf{L}_{k,u}}\mathbf{X}_k^l\|_2\ \|\mathbf{\Psi}_{k,u}^l\mathbf{\Theta}_{k,d}^l{}^\top\|_2
$$

$$(31)$$

Then, building upon (31) and for obtaining an upper bound on $E(\mathbf{X}_k^{l+1}) = \mathrm{tr}(\mathbf{X}_k^{l+1}{}^\top\mathbf{L}_{k,d}\mathbf{X}_k^{l+1}) + \mathrm{tr}(\mathbf{X}_k^{l+1}{}^\top\mathbf{L}_{k,u}\mathbf{X}_k^{l+1})$, we first elaborate on the first term:

$\mathrm{tr}(\mathbf{X}_k^{l+1}{}^\top\mathbf{L}_{k,d}\mathbf{X}_k^{l+1})$

$$
= \overbrace{\mathrm{tr}(\mathbf{\Theta}_{k,d}^l{}^\top\mathbf{X}_{k,d}^l{}^\top e^{-t_d\mathbf{L}_{k,d}}\mathbf{L}_{k,d}e^{-t_d\mathbf{L}_{k,d}}\mathbf{X}_{k,d}^l\mathbf{\Theta}_{k,d}^l)}^{\boxed{1}} + \overbrace{\mathrm{tr}(\mathbf{\Psi}_{k,d}^l{}^\top\mathbf{X}_k^l{}^\top e^{-t_d\mathbf{L}_{k,d}}\mathbf{L}_{k,d}e^{-t_d\mathbf{L}_{k,d}}\mathbf{X}_k^l\mathbf{\Psi}_{k,d}^l)}^{\boxed{2}}
$$

$$
+ \overbrace{\mathrm{tr}(\mathbf{\Psi}_{k,u}^l{}^\top\mathbf{X}_k^l{}^\top e^{-t_u\mathbf{L}_{k,u}}\mathbf{L}_{k,d}e^{-t_u\mathbf{L}_{k,u}}\mathbf{X}_k^l\mathbf{\Psi}_{k,u}^l)}^{\boxed{3}} + 2\overbrace{\mathrm{tr}(\mathbf{\Theta}_{k,d}^l{}^\top\mathbf{X}_{k,d}^l{}^\top e^{-t_d\mathbf{L}_{k,d}}\mathbf{L}_{k,d}e^{-t_d\mathbf{L}_{k,d}}\mathbf{X}_k^l\mathbf{\Psi}_{k,d}^l)}^{\boxed{4}}
$$

$$
+ 2\overbrace{\mathrm{tr}(\mathbf{\Theta}_{k,d}^l{}^\top\mathbf{X}_{k,d}^l{}^\top e^{-t_d\mathbf{L}_{k,d}}\mathbf{L}_{k,d}e^{-t_u\mathbf{L}_{k,u}}\mathbf{X}_k^l\mathbf{\Psi}_{k,u}^l)}^{\boxed{5}} + 2\overbrace{\mathrm{tr}(\mathbf{\Psi}_{k,d}^l{}^\top\mathbf{X}_k^l{}^\top e^{-t_d\mathbf{L}_{k,d}}\mathbf{L}_{k,d}e^{-t_u\mathbf{L}_{k,u}}\mathbf{X}_k^l\mathbf{\Psi}_{k,u}^l)}^{\boxed{6}}
$$

$$
\leq e^{-2t_d\lambda_{\min}^{(d)}}E_d(\overbrace{\mathbf{X}_{k,d}^l}^{\mathbf{B}_k^\top\mathbf{X}_{k-1}^l})\|\mathbf{\Theta}_{k,d}^l\|_2^2 + e^{-2t_d\lambda_{\min}^{(d)}}E_d(\mathbf{X}_k^l)\|\mathbf{\Psi}_{k,d}^l\|_2^2 + E_d(\mathbf{X}_k^l)\|\mathbf{\Psi}_{k,u}^l\|_2^2
$$

$$
+ 2.F.\lambda_{\max}^{(d)}e^{-2t_d\lambda_{\min}^{(d)}}.\|\mathbf{X}_k^l\|.\|\mathbf{X}_{k,d}^l\|.\|\mathbf{\Theta}_{k,d}^l\|.\|\mathbf{\Psi}_{k,d}^l\| + 2.F.\lambda_{\max}^{(d)}e^{-t_d\lambda_{\min}^{(d)}}.\|\mathbf{X}_k^l\|.\|\mathbf{X}_{k,d}^l\|.\|\mathbf{\Theta}_{k,d}^l\|.\|\mathbf{\Psi}_{k,d}^l\|
$$

$$
+ 2.F.\lambda_{\max}^{(d)}e^{-t_d\lambda_{\min}^{(d)}}\|\mathbf{X}_k\|^2
$$

$$
\leq s.e^{-2\varphi}\lambda_{\max}^{(u)}E_u(\mathbf{X}_{k-1}^l) + s.e^{-2\varphi}E_d(\mathbf{X}_k^l) + s.E_d(\mathbf{X}_k^l) + 2.F.s.e^{-2\varphi}.\lambda_{\max}^{(d)}.\|\mathbf{X}_k^l\|.\|\mathbf{X}_{k,d}^l\|
$$

$$
+ 2.F.s.e^{-\varphi}.\lambda_{\max}^{(d)}.\|\mathbf{X}_k^l\|.\|\mathbf{X}_{k,d}^l\| + 2.F.s.e^{-\varphi}.\lambda_{\max}^{(d)}.\|\mathbf{X}_k^l\|^2
$$

$$
\leq s.e^{-2\varphi}\tilde{\lambda}_{\max}E_u(\mathbf{X}_{k-1}^l) + s.(e^{-2\varphi}+1).E_d(\mathbf{X}_k^l) + 2.F.s.(e^{-\varphi}+e^{-2\varphi}).\tilde{\lambda}_{\max}^{1.5}.\|\mathbf{X}_k^l\|.\|\mathbf{X}_{k-1}^l\| + 2.F.s.e^{-\varphi}.\tilde{\lambda}_{\max}.\|\mathbf{X}_k^l\|^2
$$

$$(32)$$

and similarly for the second term

$\mathrm{tr}(\mathbf{X}_k^{l+1}{}^\top\mathbf{L}_{k,u}\mathbf{X}_k^{l+1})$

$$
\leq s.e^{-2\varphi}\tilde{\lambda}_{\max}E_d(\mathbf{X}_{k+1}^l) + s.(e^{-2\varphi}+1).E_u(\mathbf{X}_k^l) + 2.F.s.(e^{-\varphi}+e^{-2\varphi}).\tilde{\lambda}_{\max}^{1.5}.\|\mathbf{X}_k^l\|.\|\mathbf{X}_{k+1}^l\| + 2.F.s.e^{-\varphi}.\tilde{\lambda}_{\max}.\|\mathbf{X}_k^l\|^2.
$$

$$(33)$$

Therefore, the final upper bound on $E(\mathbf{X}_k^{l+1})$ can be expressed in a combined form as

$$
E(\mathbf{X}_k^{l+1}) \leq s.e^{-2\varphi}\tilde{\lambda}_{\max}(E(\mathbf{X}_{k-1}^l) + E(\mathbf{X}_{k+1}^l))
$$
$$
+ s.(e^{-2\varphi}+1)E(\mathbf{X}_k^l) + 2Fs.(e^{-\varphi}+e^{-2\varphi})\tilde{\lambda}_{\max}^{1.5}\|\mathbf{X}_k^l\|(\|\mathbf{X}_{k-1}^l\|+\|\mathbf{X}_{k+1}^l\|) + 2Fs.e^{-\varphi}\tilde{\lambda}_{\max}\|\mathbf{X}_k^l\|^2
$$

$$(34)$$

where

$$
\varphi := \min_k\{t_d\lambda_{\min}(\mathbf{L}_{k,d}), t_u\lambda_{\min}(\mathbf{L}_{k,u})\}
$$

$$
\tilde{\lambda}_{\max} := \max_k\{\lambda_{\max}(\mathbf{L}_{k,d}), \lambda_{\max}(\mathbf{L}_{k,u})\}
$$

$$
s := \sqrt{\max_{k,l}\{\|\mathbf{\Theta}_{k,d}^l\|, \|\mathbf{\Theta}_{k,u}^l\|, \|\mathbf{\Psi}_{k,d}^l\|, \|\mathbf{\Psi}_{k,u}^l\|\}}.
$$

$$(35)$$

Therefore, the proof is completed. □

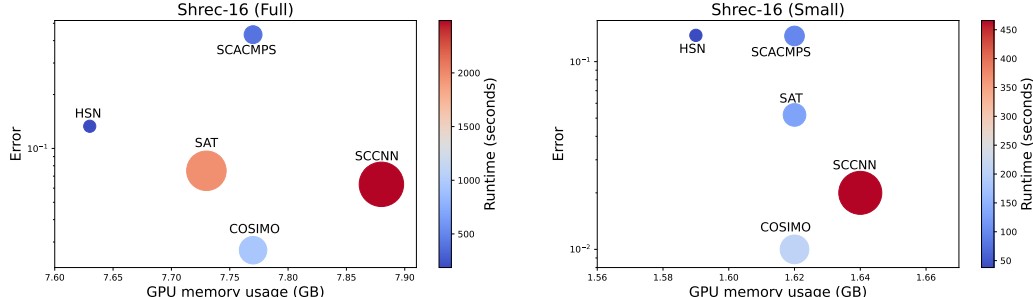

Figure 5: Comparison of the performance error and GPU memory usage (in GB) across runtime (in seconds) (in both color and circle size) on both the Small and Full versions of the `Shrec-16` dataset. The proposed COSIMO method has a good trade-off between runtime and memory usage while performing considerably better.

## G  Proof of Corollary 5.4

It follows similar to the justifications mentioned in Section E for the second and third terms of the results of Theorem 5.3.

## H  Proof of Proposition 5.5

*Proof.* We start from $\ln(s) > 2\varphi - \ln(\tilde{\lambda}_{\max})$. By considering the fact that $\lambda_{\min}(\mathbf{L}_{k,d}) \leq \min(\lambda_{\min}(\mathbf{L}_{k,u}), \lambda_{\min}(\mathbf{L}_k))$, assuming $t_d = t_u = t$, replacing $\varphi = t\lambda_{\min}(\mathbf{L}_k)$, and the definition $k_f(\mathbf{L}_k) := \frac{\lambda_{\max}(\mathbf{L}_k)}{\lambda_{\min}(\mathbf{L}_k)}$ (with $\lambda_{\min}(\mathbf{L}_k) \neq 0$) [39], one can write:

$$\ln(s) > 2\varphi - \ln(\tilde{\lambda}_{\max}) \rightarrow t < \frac{\ln(s\tilde{\lambda}_{\max})}{2\lambda_{\min}(\mathbf{L}_k)} < \frac{\ln(s\tilde{\lambda}_{\max}) + 2\lambda_{\max}(\mathbf{L}_k)}{2\lambda_{\min}(\mathbf{L}_k)} < \frac{\ln(s\tilde{\lambda}_{\max})}{2\lambda_{\min}(\mathbf{L}_k)} + k_f(\mathbf{L}_k).$$
(36)

$\square$

## I  Runtime and Memory Usage Comparison in the Performance Trade-off

We provide additional comparison results of runtime (in seconds) and memory usage (in GB) on both Small and Full versions of the `Shrec-16` dataset compared to the SOTA in Fig. 5. In general and as observed from these results, the proposed COSIMO method enjoys a trade-off between runtime and memory usage while performing considerably better. More precisely, the proposed COSIMO method is ranked in the middle of the SOTA runtime and memory usage while providing significantly better accuracy performance. Note that, in COSIMO these metrics are heavily relying on the number of selected eigenvalue-eigenvector pairs in simplicial subspaces and therefore it can be optimized in a data-driven manner.

## J  Permutation Equivariance Property of COSIMO

**Property (Permutation equivariance [26]).** Consider a simplicial complex $\mathcal{X}$ characterized by boundary operators $\mathcal{B} = \{\mathbf{B}_k\}_{k=1}^{K}$. Let $\mathcal{P} = \{\mathbf{P}_k\}_{k=0}^{K}$ represent a sequence of permutation matrices, where each $\mathbf{P}_k$ is of size $|\mathcal{X}_k| \times |\mathcal{X}_k|$ and corresponds to the chain complex dimensions $\{C_k\}_{k=0}^{K}$, ensuring $\mathbf{P}_k \in \mathbb{R}^{|\mathcal{X}_k| \times |\mathcal{X}_k|}$. We define the permuted boundary operator as

$$[PB]_k := \mathbf{P}_{k-1}\mathbf{B}_k\mathbf{P}_k^{\top}.$$

A simplicial convolutional network (SCN) with the learnable weight matrix $\mathbf{W}$ is said to be permutation equivariant if, for any such transformation $\mathbf{P}$, the following holds:

$$\mathrm{SCN}_{\mathbf{W},\mathbf{B}}(\mathbf{c}_j) = \mathbf{P}_{\ell}\,\mathrm{SCN}_{\mathbf{W},PB}(\mathbf{P}_j\mathbf{c}_j).$$
(37)

Based on the above-mentioned properties, we show that COSIMO governs them in the following proposition.

**Proposition J.1.** *The COSIMO model stated in* (9) *exhibits the property of Permutation Equivariance.*

*Proof.* First, by considering $P\mathbf{L}_{k,d} = (\mathbf{P}_{k-1}\mathbf{B}_k\mathbf{P}_k)^\top(\mathbf{P}_{k-1}\mathbf{B}_k\mathbf{P}_k) = \mathbf{P}_k^\top \mathbf{L}_{k,d}\mathbf{P}_k$ and similarly $P\mathbf{L}_{k,u} = \mathbf{P}_k^\top \mathbf{L}_{k,u}\mathbf{P}_k$, and also $P\mathbf{X}_{k,d} = (\mathbf{P}_{k-1}\mathbf{B}_k\mathbf{P}_k^\top)^\top(\mathbf{P}_{k-1}\mathbf{X}_{k-1}) = \mathbf{P}_k\mathbf{B}_k^\top \mathbf{X}_{k-1} = \mathbf{P}_k\mathbf{X}_{k,d}$ and similarly $P\mathbf{X}_{k,u} = \mathbf{P}_k\mathbf{X}_{k,u}$, the permuted exponential expansion can be written as follows:

$$
\begin{aligned}
&\mathbf{P}_k\,\mathrm{COSIMO}_{\mathbf{W},PB}(\{\mathbf{P}_{k-1}\mathbf{c}_{k-1}, \mathbf{P}_k\mathbf{c}_k, \mathbf{P}_{k+1}\mathbf{c}_{k+1}\}) \\
&= \mathbf{P}_k\sigma\left(\mathbf{P}_k^\top e^{-t_d\mathbf{L}_{k,d}}\mathbf{X}_{k,d}^{l-1}\mathbf{\Theta}_{k,d}^l + \mathbf{P}_k^\top e^{-t_u\mathbf{L}_{k,u}}\mathbf{X}_{k,u}^{l-1}\mathbf{\Theta}_{k,u}^l + \mathbf{P}_k^\top e^{-t_d\mathbf{L}_{k,d}}\mathbf{X}_k^{l-1}\mathbf{\Psi}_{k,d}^l\right. \\
&\quad \left.+\mathbf{P}_k^\top e^{-t_u\mathbf{L}_{k,u}}\mathbf{X}_k^{l-1}\mathbf{\Psi}_{k,u}^l\right) \\
&= \sigma\left(e^{-t_d\mathbf{L}_{k,d}}\mathbf{X}_{k,d}^{l-1}\mathbf{\Theta}_{k,d}^l + e^{-t_u\mathbf{L}_{k,u}}\mathbf{X}_{k,u}^{l-1}\mathbf{\Theta}_{k,u}^l + e^{-t_d\mathbf{L}_{k,d}}\mathbf{X}_k^{l-1}\mathbf{\Psi}_{k,d}^l + e^{-t_u\mathbf{L}_{k,u}}\mathbf{X}_k^{l-1}\mathbf{\Psi}_{k,u}^l\right) \\
&= \mathrm{COSIMO}_{\mathbf{W},\mathbf{B}}(\{\mathbf{c}_{k-1}, \mathbf{c}_k, \mathbf{c}_{k+1}\}),
\end{aligned}
\tag{38}
$$

which completes the proof. $\square$

# K  More Details on the Datasets

**Trajectory prediction.** It is important to note that trajectory prediction in this context involves identifying a candidate node within the neighborhood of the target node, a process influenced by node degree. Given that the average node degree is $5.24$ in the synthetic dataset and $4.81$ in the `ocean-drifts` dataset, a random guess would achieve approximately $20\%$ accuracy. The high standard deviations observed, particularly in the `ocean-drifts` dataset, may be attributed to its limited size. For more information, please refer to [26].

**Regression on partial deformable shapes.** The sampling of the shapes were in regular cuts, where template shapes were sliced at six orientations, producing 320 partial shapes, and irregular holes, where surface erosion was applied based on area budgets ($40\%$, $70\%$, and $90\%$), yielding 279 shapes. This creates to a dataset of 599 shapes across eight classes (humans and animals), with varying missing areas ($10\%$–$60\%$).

**Node classification.** To provide more details about the NC datasets, we have outlined their statistics in Table 4.

# L  Implementation Details

In certain cases, we mostly use TopoModelX [57, 58, 24] to implement previous SOTA methods. For accessing and processing real-world datasets, we employ Torch TopoNetX [58]. For the experiments on trajectory prediction, we use the aggregation of $M$ branches discussed in Remark 4.3. We use cross-validation for tuning the possible hyperparameters with the selected values provided in Table 5. Detailed hyperparameter configurations for both synthetic and real-world datasets are provided in the code in https://github.com/ArefEinizade2/COSIMO. For experimental results on `synthetic` and `ocean-drifts`, we followed the experimental settings from reference papers [26, 18]. The experiments were conducted on an A100 NVIDIA GPU with 40 GB of memory.

**Number of selected eigenvalue-eigenvector $K$.** In our framework, the selection of an appropriate $K$ can be approached in two ways: (1) supervised and (2) unsupervised. In the supervised setting — which serves as the primary approach in this work — $K$ is determined through cross-validation over a reasonable range of values to identify the configuration yielding the best performance. For the unsupervised case, alternative strategies can be employed. For example, we explore the use of spectral entropy [59] as a proof of concept to assess its potential effectiveness, which is defined as follows:

$$
H := -\sum_{i=1}^{n} p_i \log p_i; \quad \text{where} \quad p_i = \frac{\lambda_i}{\sum_j \lambda_j}.
\tag{39}
$$

Table 4: Node classification dataset statistics.

| Dataset | Simplex | # 0-simplicies | # 1-simplicies | # 2-simplicies | Order |
|---|---|---|---|---|---|
| high-school | Group of people | 327 | 5818 | 2370 | 3 |
| senate-bills | Co-sponsors | 294 | 6974 | 3013 | 3 |

Table 5: Hyperparameter details for each dataset; $lr$ and $n_{\text{epochs}}$ are the learning rate and number of epochs, respectively.

| Hyperparam | synthetic | ocean-drifts | Shrec-small | Shrec-full | high-school | senate-bills | proteins |
|---|---|---|---|---|---|---|---|
| $lr$ | $5 \times 10^{-3}$ | $5 \times 10^{-2}$ | $10^{-2}$ | $10^{-2}$ | $10^{-3}$ | $10^{-2}$ | $10^{-3}$ |
| Optimizer | ADAM | ADAM | ADAM | ADAM | ADAM | ADAM | ADAM |
| Batch size | 100 | 100 | 256 | 512 | 256 | 256 | 256 |
| $n_{\text{epochs}}$ | 1000 | 1000 | 100 | 100 | 700 | 100 | 30 |
| $M$ | 3 | 3 | 1 | 1 | 1 | 1 | 1 |

Table 6: Node classification accuracy across selected optimal value of eigenvalue-eigenvector ($K$) pairs by unsupervised (Spectral entropy) and supervised (cross-validation) methods on the high-school dataset.

|  | Spectral entropy | Cross validation |
|---|---|---|
| $K$ | 14 | 10 |
| Acc (%) | 92.0 | 91.9 |

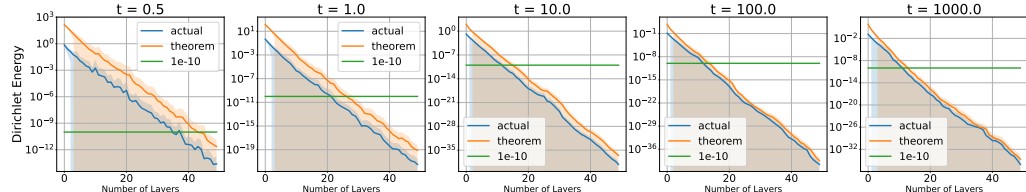

Figure 6: Over-smoothing results of COSIMO across different layer depths. Each subplot corresponds to a different parameter $t$ in COSIMO, showing the evolution of the Dirichlet energy as a function of the number of layers.

One can choose $K$ where the entropy contribution of the next eigenvalues becomes negligible.

We experimentally apply supervised and unsupervised methods for the selection of $K$ on the high-school node classification dataset in Table 6. Note that the cross-validation method selects $\sim 3\%$ of eigenvalues for $\mathbf{L}_0$, which is fairly consistent with the spectral entropy methodology ($\sim 4.5\%$).

## M  A Deeper Look at Over-Smoothing

### M.1  The effect of varying the Hodge receptive field $t$

To study the effect of varying the Hodge receptive field $t$ on the effective number of layers, we first generate 100 random realizations of simplices with some random (filled) holes in them using the approach introduced in [26]. Then, by fixing the number of hidden features to 4, we varied the number of layers and monitored the actual and theoretical bounds in Theorem 5.3 averaged over the random realizations. The results are shown in Fig. 6. We considered a threshold of $10^{-10}$ as the over-smoothing occurrence threshold. As observed, increasing $t$ reduces the effective number of layers from approximately 40 to 15. Besides, the difference between the actual and theoretical bounds gradually decreases and approaches zero, demonstrating the descriptive results of Theorem 5.3.

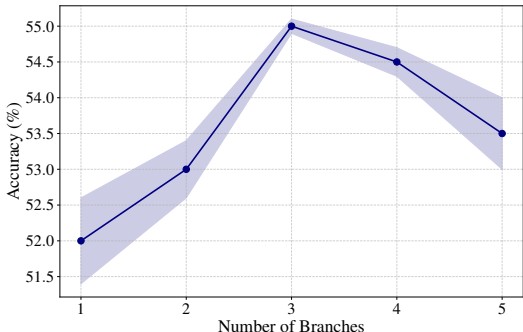

Figure 7: Sensitivity analysis on the number branches ($M$) of COSIMO.

Note that decreasing $t$ too much can negatively impact the topology of the simplicial complex, potentially leading to issues like over-squashing of information [60], degrading performance. A thorough analysis of this variation is beyond the scope of the paper and is explored in future work.

## M.2 Connection with related work [18]

If we consider a simplified filtering framework as $\mathbf{X}_k^{l+1} = (\mathbf{I} - \mathbf{L}_k)\mathbf{X}_k^l\mathbf{W}^l + \mathbf{X}_{k,d}^l\mathbf{W}_d^l + \mathbf{X}_{k,u}^l\mathbf{W}_u^l$ in (4), for $E(\mathbf{X}_k^{l+1}) = \mathrm{tr}(\mathbf{X}_k^{l+1^\top}\mathbf{L}_{k,d}\mathbf{X}_k^{l+1}) + \mathrm{tr}(\mathbf{X}_k^{l+1^\top}\mathbf{L}_{k,u}\mathbf{X}_k^{l+1})$, we can elaborate on the first term as:

$$
\begin{aligned}
\mathrm{tr}(\mathbf{X}_k^{l+1^\top}\mathbf{L}_{k,d}\mathbf{X}_k^{l+1}) &= \mathrm{tr}([\mathbf{W}^{l^\top}\mathbf{X}_k^{l^\top}(\mathbf{I} - \mathbf{L}_{k,d})]\mathbf{L}_{k,d}[(\mathbf{I} - \mathbf{L}_{k,d})\mathbf{X}_k^l\mathbf{W}^l]) \\
&+ \mathrm{tr}(\mathbf{W}_d^{l^\top}\mathbf{X}_{k,d}^{l^\top}\mathbf{L}_{k,d}\mathbf{X}_{k,d}^l\mathbf{W}_d^l) + 2\,\mathrm{tr}([\mathbf{W}^{l^\top}\mathbf{X}_k^{l^\top}(\mathbf{I} - \mathbf{L}_{k,d})]\mathbf{L}_{k,d}\mathbf{X}_{k,d}^l\mathbf{W}_d^l) \\
&\leq s.\lambda_{max}^2(\mathbf{I} - \mathbf{L}_{k,d})E_d(\mathbf{X}_k^l) + F.s.\lambda_{max}(\mathbf{L}_{k,d}).\|\mathbf{X}_{k,d}^l\|^2 \\
&+ 2.F.s.\lambda_{max}(\mathbf{I} - \mathbf{L}_{k,d}).\lambda_{max}(\mathbf{L}_{k,d}).\|\mathbf{X}_k^l\|.\|\mathbf{X}_{k,d}^l\|.
\end{aligned}
\tag{40}
$$

Similarly for the second term, we have:

$$
\begin{aligned}
\mathrm{tr}(\mathbf{X}_k^{l+1^\top}&\mathbf{L}_{k,u}\mathbf{X}_k^{l+1}) \\
&\leq s.\lambda_{max}^2(\mathbf{I} - \mathbf{L}_{k,u})E_u(\mathbf{X}_k^l) + F.s.\lambda_{max}(\mathbf{L}_{k,u}).\|\mathbf{X}_{k,u}^l\|^2 \\
&+ 2.F.s.\lambda_{max}(\mathbf{I} - \mathbf{L}_{k,u}).\lambda_{max}(\mathbf{L}_{k,u}).\|\mathbf{X}_k^l\|.\|\mathbf{X}_{k,u}^l\|.
\end{aligned}
\tag{41}
$$

So, in a combined manner, we can write:

$$
\begin{aligned}
E(\mathbf{X}_k^{l+1})& \\
&\leq s.\lambda_{max}^2(\mathbf{I} - \mathbf{L}_k)E(\mathbf{X}_k^l) + F.s.\lambda_{max}(\mathbf{L}_{k,d}).\|\mathbf{X}_{k,d}^l\|^2 + F.s.\lambda_{max}(\mathbf{L}_{k,u}).\|\mathbf{X}_{k,u}^l\|^2 \\
&+ 2.F.s.\lambda_{max}(\mathbf{I} - \mathbf{L}_k).\lambda_{max}(\mathbf{L}_k).\|\mathbf{X}_k^l\|.(\|\mathbf{X}_{k,d}^l\| + \|\mathbf{X}_{k,u}^l\|).
\end{aligned}
\tag{42}
$$

If we consider the simplified framework with $F = 1$ as $\mathbf{x}_k^{l+1} = w_0(\mathbf{I} - \mathbf{L}_k)\mathbf{x}_k^l + w_1\mathbf{x}_{k,d} + w_2\mathbf{x}_{k,u}$, which was proposed in [31], the bound in (43) takes the form of:

$$
\begin{aligned}
E(\mathbf{x}_k^{l+1})& \\
&\leq s.\lambda_{max}^2(\mathbf{I} - \mathbf{L}_k)E(\mathbf{x}_k^l) + s.\lambda_{max}(\mathbf{L}_{k,d}).\|\mathbf{x}_{k,d}^l\|^2 + s.\lambda_{max}(\mathbf{L}_{k,u}).\|\mathbf{x}_{k,u}^l\|^2 \\
&+ 2.s.\lambda_{max}(\mathbf{I} - \mathbf{L}_k).\lambda_{max}(\mathbf{L}_k).\|\mathbf{x}_k^l\|.(\|\mathbf{x}_{k,d}^l\| + \|\mathbf{x}_{k,u}^l\|),
\end{aligned}
\tag{43}
$$

which partially coincides with the over-smoothing bound obtained in [18], with an additional fourth term missing in the previous work [18].

## N   Sensitivity on the Number of Branches ($M$) of COSIMO

Figure 7 depicts the accuracy results on the Ocean trajectory prediction task for different values of $M$. It is observed that increasing $M$ up to 3 enhances the expressivity of COSIMO, as stated in Remark 4.3. However, continuing to increase beyond 3 results in performance degradation, likely due to overfitting.

Table 7: Experimental validation of the stability Theorem 4.4: the practical gap between left and right sides in (13) in Theorem 4.4.

|  | $SNR_1 = -5$ | $SNR_1 = 0$ | $SNR_1 = 10$ | $SNR_1 = 20$ |
|---|---|---|---|---|
| $SNR_2 = -5$ | 12.63 | 9.55 | 8.74 | 8.62 |
| $SNR_2 = 0$ | 5.72 | 2.71 | 1.88 | 1.79 |
| $SNR_2 = 10$ | 4.11 | 1.13 | 0.31 | 0.18 |
| $SNR_2 = 20$ | 3.86 | 0.91 | 0.11 | 0.01 |

## O   Experimental Validation of the Stability Theorem 4.4

Following the SNR setting of Figure 4, Table 7 shows the difference between the right and left bounds in Theorem 4.4 in (13). We observe that, as the $SNR_1$ and $SNR_2$ increase, the gap tends to get tighter.

## P   Broader Impact

In the paper, we have shown the applicability of the proposed approach, *i.e.*, COSIMO, on a wide range of real-world societal applications, including ocean drift prediction, regression of the uncompleted mesh shapes, social data analysis, and protein structure classification. This can illustrate the societal impact of the COSIMO for improvement in different humanistic aspects like sea navigation, computer vision, social network development, and medical informatics.

