# OpenReview forum: "Continuous Simplicial Neural Networks"
_NeurIPS.cc/2025/Conference — NeurIPS 2025 poster_

### Official Review · Reviewer_z9BE · 2025-06-27

**Clarity:** 3
**Significance:** 3
**Originality:** 3
**Rating:** 5
**Confidence:** 3

**Summary:**

The authors develop a continuous-time formulation for training neural networks on simplical networks using discrete PDE theory. They show theoretically that the method prevents over-smoothing and is stable to perturbations to the simplex structure. The method is compared against a wide range of baselines and improves performance across multiple benchmarks.

**Questions:**

- Theorem 4.3: the proof does not seem to take into account that B_k is an integer matrix. This would mean ||E_k|| is always at least 1, which breaks the “eps_k is small” assumption in eq 19.
- Theorem 4.3: it is not clear to me why being robust to perturbations of B_k is a useful property. Could you provide an example of a practical scenario where such perturbations might appear?
- Section 6.3: The analysis of over-smoothing is limited to randomly initialized networks. It is therefore not clear how this phenomenon impacts the performance of trained networks. In principle, the network could learn to compensate for the initial over-smoothing during the training process. To complete the analysis, it would be interesting to repeat at least one of the experiments in section 6.1 for various settings ot t, to see if the “over-smoothed” variants actually have degraded performance.
- what value of t was used for the experiments?
- Is eq 8 is a definition or does it follow directly from eq 7. Please provide a proof/justification.

**Ethical Concerns:**

["NO or VERY MINOR ethics concerns only"]

**Final Justification:**

The authors have addressed all my comments and provided additional experiments confirming the theoretical findings.

**Limitations:**

yes

**Paper Formatting Concerns:**

no formatting concerns

**Quality:**

3

**Strengths And Weaknesses:**

Strengths:
- Making use of the simplical structure with learning based method is an important problem, and I am glad to see research in this direction.
- The method outperforms previous state of the art SCCNN, while also reducing runtime and GPU memory

Weaknesses:
- The connection between theorems the theorems and practical implications are not always clearly explained (see questions)

---

> ### Author Rebuttal · Authors · 2025-07-29
>
> We appreciate the reviewer for providing constructive comments on our paper. We'll include the feedback in the camera-ready paper. In the following, we have addressed the reviewer's concerns:
>
> **Q1:** Firstly, our framework and its theoretical claims are also valid and straightforwardly transferable to the case of weighted structures, not just binarized. Therefore, the incidence matrices can be non-integer in general. Also, the proof of Theorem 4.3 does not need the error matrices to have small norm. Theorem 4.3 holds in general, and having a small norm is only needed in the proof of Corollary 4.4. We will clarify these points in the camera-ready.
>
> We'd also like to clarify the meaning of having a small norm for $\mathbf{E}_k$. From the proof of Corollary 4.4, we observe that being small is *w.r.t* the scale of the maximum eigenvalue of the Hodge Laplacians, *i.e.*, $\lambda\_{max}(\mathbf{L}\_{k,d})$. Therefore, it is not necessarily considered as being small in an absolute manner.
>
> Finally, regarding the nature of error matrices $\mathbf{E}_k$, we clarify that the removal, addition, and weight changing of edges can be modeled by the additive model $\hat{\mathbf{B}}_1=\mathbf{B}_1+\mathbf{E}_1$.
>
> To simplify the discussion, we start with a simple toy example. Let's consider an undirected three-node graph in which $\mathcal{V}=${$1,2,3$} and $\mathcal{E}=${$(1,2),(1,3)$}. In this case, we have $\mathbf{B}_1=[[1 \\:\\:\\: 1][-1 \\:\\:\\: 0][0 \\:\\:\\: -1]]\in\mathbb{R}^{3\times2}$. Then, one special form of an incidence error matrix corresponding to changes in the current edge weights can be stated as $\mathbf{E}\_1=[[-\epsilon\_{12} \\:\\:\\: -\epsilon\_{13}][\epsilon\_{12} \\:\\:\\: 0][0 \\:\\:\\: \epsilon\_{13}]]\in\mathbb{R}^{3\times2}$. Therefore, after structural perturbation, we have $\hat{\mathbf{B}}_1=\mathbf{B}_1+\mathbf{E}_1=[[1-\epsilon\_{12} \\:\\:\\: 1-\epsilon\_{13}][-1+\epsilon\_{12} \\:\\:\\: 0][0 \\:\\:\\: -1+\epsilon\_{13}]]\in\mathbb{R}^{3\times2}$. So, we have $\\|\mathbf{E}\_1\\|=\sqrt{2}\sqrt{\epsilon^2\_{12}+\epsilon^2\_{13}}$, and, given the scale of $\epsilon\_{12}$ and $\epsilon\_{13}$, this norm can be small or not. As a conclusion, since our definition of simplicial perturbation is not limited to only the addition or removal of edges (or higher-order connections), the norm of the error matrices can take different scales. On the other hand, and in the case of weighted graphs and in connection with the previously mentioned examples, assume that we have the edge weight $e\_{12}=0.0625$ (instead of 1). This implies that $\mathbf{B}_1=[[0.25 \\:\\:\\: 1][-0.25 \\:\\:\\: 0][0 \\:\\:\\: -1]]\in\mathbb{R}^{3\times2}$. Now, for only modeling the removal of this edge, we just need to have $\epsilon\_{12}=0.25$ in $\mathbf{E}_1$, leading to a small norm for $\mathbf{E}\_1$. Therefore, the scale of the norm for $\mathbf{E}$ heavily depends on the scale of edge weights.
>
> **Q2:** In connection with the response to the previous question, the perturbation of $\mathbf{B}_k$ could model potential erroneous structures where we have the addition of non-existing simplicial connections, the removal of existing simplices, or changes in simplicial connection weights in a principled manner. For further clarification, one can state the toy graph from the response to **Q1** as $\mathcal{V}=${$1,2,3$} and $\mathcal{E}=${$(1,2),(1,3),(2,3)$} in which edge $(2,3)$ takes zero value because of disconnectivity. Then, the incidence matrix is stated as $\mathbf{B}_1=[[1 \\:\\:\\: 1 \\:\\:\\: 0][-1 \\:\\:\\: 0 \\:\\:\\: 0][0 \\:\\:\\: -1 \\:\\:\\: 0]]\in\mathbb{R}^{3\times3}$. Now, as an example, for modeling removing edge $(1,2)$ and adding edge $(2,3)$, the error matrix can take the form of $\mathbf{E}_1=[[-1\\:\\:\\: 0 \\:\\:\\: 0][1 \\:\\:\\: 0 \\:\\:\\: 1][0 \\:\\:\\: 0 \\:\\:\\: -1]]\in\mathbb{R}^{3\times3}$. Again, if we consider the case of changing edge weights (not removal or addition) or a weighted graph, the norm of the error matrix can take a very small value. Therefore, the stability of COSIMO against perturbations shows its robustness to such cases.
>
> **Q3:** We'll clarify in the answer to **Q4** that $t$ is indeed a parameter of the model, not a hyperparameter. Previous studies typically analyze the over-smoothing phenomenon by stacking several graph or simplicial layers. Therefore, we provide comprehensive results on the $\texttt{high-school}$ node classification dataset averaged over three random seeds in the following table regarding the tradeoff between over-smoothing and accuracy. We observe that the proposed COSIMO can maintain the performance even with an increasing number of layers up to 16 in terms of accuracy and Dirichlet energy. We also observe that COSIMO is more robust against over-smoothing than SCCNN, where extremely smoothness happens after $n_l=4$ and leads to severely reduced performance. The key observation and trend here are also fairly consistent with the results in Figure 3.
>
> **Acc:**
> |**Method** | $n_l$=1 | $n_l$=2 | $n_l$=4 | $n_l$=8 | $n_l$=16 | $n_l$=32|
> |---|---|---|---|---|---|---|
> |SCCNN | 57 | 81 | 43 | 16 | 12 | 10 |
> |COSIMO | 79.33 | 86.33 | 91.33 | 91.33 | 88.00 | 53.67|
>
> **Dirichlet Energy:**
> |**Method** | $n_l$=1 | $n_l$=2 | $n_l$=4 | $n_l$=8 | $n_l$=16 | $n_l$=32|
> |---|---|---|---|---|---|---|
> |SCCNN | 0.88| 0.80| 0.75| 0.53 | 0.42 | 0.07|
> |COSIMO | 0.94 | 0.92 | 0.92 | 0.92 | 0.88 | 0.19|
>
> **Q4:** As stated in Remark 5.6, and due to the differentiability of our framework *w.r.t.* the simplicial receptive fields, in all the experimental results (except the over-smoothing analysis in Section 6.2), the values of $t$ are learned during the training process, *i.e.*, $t$ are parameters, not hyperparameters. We considered $t$ as a hyperparameter only in the over-smoothing analysis in Section 6.2 to illustrate the effect of different values for $t$ on the over-smoothing phenomenon.
>
> **Q5:** Eq. (8) implies a definition. In fact, it integrates joint and independent dynamics that act as source terms; this has been shown to improve robustness to over-smoothing and stability as outlined in [1]. The analytical form of the solution to the set of PDEs in eqs. (5), (6), and (7) take the form of eq. (9), which can be stated in a **general form** as eq. (8). Note that $\mathbf{x}\_{k,d}(t\_d) + \mathbf{x}\_{k,u}(t\_u)$ in eq.(8) is actually the general symbolic form of $\mathbf{x}\_{k}(t\_d,t\_u)$ in eq. (9). We will modify and clarify these points in the camera-ready version.
>
> ---
> References:
>
> [1] A. Han, et al, “From continuous dynamics to graph neural networks: Neural diffusion and beyond,” TMLR 2025

---

> > ### Comment · Reviewer_z9BE · 2025-08-03
> >
> > I thank the authors for their detailed response. Especially for the additional experiments for Q3, which confirm that over-smoothing is indeed also reduced in trained models.
> >
> > My concerns are mostly addressed. Considering Q1: I would be interested to know how  $\lambda_{max}(\mathbf{L}_{k,d})$ compares to $\|\|E_k\|\|$ in practice. For example, would it be possible to compare the theoretical bound from Thm 4.3 with the experimental results in Figure 4?

---

> > > ### Author Response · Authors · 2025-08-04
> > > **More clarification for Question 1:**
> > >
> > > We again thank the reviewer for providing constructive comments. We have addressed the remaining question as follows:
> > >
> > > As required, following the SNR setting of Figure 4, we have provided the difference between the Right and Left bounds in Theorem 4.3, *i.e.*, eq. (13), in the following table. We observe that, as the SNR$_1$ and SNR$_2$ increase, the gap tends to get tighter and tighter, showcasing the applicability of Theorem 4.3 for describing the stability characteristics of the proposed framework.
> > >
> > > **The practical gap between the Left and Right sides eq. (13) in Theorem 4.3:**
> > > | | SNR$_1=-5$ | SNR$_1=0$ | SNR$_1=10$ | SNR$_1=20$|
> > > |---|---|---|---|---|
> > > |SNR$_2=-5$ | 12.63 | 9.55 | 8.74 | 8.62|
> > > |SNR$_2=0$ | 5.72 | 2.71 | 1.88 | 1.79|
> > > |SNR$_2=10$ | 4.11 | 1.13 | 0.31 | 0.18|
> > > |SNR$_2=20$ | 3.86 | 0.91 | 0.11 | 0.01|
> > >
> > > This interpretation is also confirmed by the results in the following table, which illustrates the measures of $\frac{\lambda\_{max}(\mathbf{L}\_{k,d})}{\epsilon\_k}$ and $\frac{\lambda\_{max}(\mathbf{L}\_{k,u})}{\epsilon\_{k+1}}$ across different SNRs. From this table, we observe that the scale of $\epsilon\_k$ (and $\epsilon\_{k+1}$) is getting fairly negligible compared to $\lambda\_{max}(\mathbf{L}\_{k,d})$ (and $\lambda\_{max}(\mathbf{L}\_{k,u})$) by increasing SNR (from $\sim$SNR$=10$). Therefore, as a conclusion, apart from experimental description of the stability by general Theorem 4.3, Corollary 4.4 can be applied to roughly moderate or low noisy settings, *i.e.*, SNR$=10$ to SNR$=20$, validating its practical use cases.
> > >
> > > **The measures of $\frac{\lambda\_{max}(\mathbf{L}\_{k,d})}{\epsilon\_k}$ and $\frac{\lambda\_{max}(\mathbf{L}\_{k,u})}{\epsilon\_{k+1}}$ across different SNRs:**
> > >
> > > | | SNR$=-5$ | SNR$=0$ | SNR$=10$ | SNR$=20$|
> > > |---|---|---|---|---|
> > > |$\frac{\lambda\_{max}(\mathbf{L}\_{k,d})}{\epsilon\_k}$ | 2.00 | 3.56 | 11.18 | 35.38|
> > > |$\frac{\lambda\_{max}(\mathbf{L}\_{k,u})}{\epsilon\_{k+1}}$ | 1.81 | 3.21 | 10.18 | 32.28|

---

> > > > ### Comment · Reviewer_z9BE · 2025-08-05
> > > >
> > > > I thank the authors for the additional results, confirming the validity of Theorem 4.3. With this and the previous response in mind, I will raise my score to accept.

---

### Official Review · Reviewer_tWza · 2025-07-01

**Clarity:** 4
**Significance:** 3
**Originality:** 3
**Rating:** 5
**Confidence:** 2

**Summary:**

This paper presents a continuous simplicial neural network (SNN) architecture, COSIMO, derived from partial differential equations (PDEs) on simplicial complexes. Rather than relying on fixed, discrete polynomial filters of the Hodge Laplacians, COSIMO models feature propagation as coupled heat‐diffusion processes over the lower and upper Laplacians, yielding dynamic receptive fields that adapt continuously to the underlying topology. Each COSIMO layer applies exponential Hodge filters—computed via eigen‐decompositions—followed by learnable linear projections and nonlinearities. The authors prove that COSIMO is stable under simplicial perturbations and provides formal control over over-smoothing, and they demonstrate its effectiveness on trajectory prediction, partial mesh regression, and node/graph classification, where it outperforms state-of-the-art discrete SNNs and GNNs.

**Questions:**

How do you choose the number of eigenpairs K for each Hodge Laplacian in practice? Have you evaluated accuracy versus cost trade‐offs as K varies?

**Ethical Concerns:**

["NO or VERY MINOR ethics concerns only"]

**Final Justification:**

Since my only concern that applying eigen-decomposition may consume the majority part of the computational efficiency is addressed, I will raise the score. Meanwhile, I am not an expert in this field, my confidence will remain low.

**Limitations:**

Yes

**Quality:**

4

**Strengths And Weaknesses:**

### Strengths
- Cast simplical complexes into the PDEs framework and provide allows continuous information flow between structural data.
- Bring theoretical analysis and proof to ensure the stability of the proposed work and explicit bound on over-smoothing rate.
- Different types of tasks demonstrates the effectiveness of this method
### Weakness
The eigenvalue decomposition may hinder the actual applicability of this method

---

> ### Author Rebuttal · Authors · 2025-07-29
>
> We appreciate the reviewer for providing constructive comments on our paper. We'll include the feedback in the camera-ready paper. In the following, we have addressed the reviewer's concerns:
>
> **W1 and Q1:** First, we emphasize that performing EVD on the Hodge Laplacians is done once and as a preprocessing step, so they can be precomputed. Additionally, as will be presented in the following, one can rely on a small subset of eigenvalue-eigenvector pairs, and then, the computational complexity of performing EVD is reduced from $\mathcal{O}(N^3)$ to $\mathcal{O}(N^2K)$. In our framework, one can select a proper $K$ using supervised (cross-validation) or unsupervised methodologies. We used the cross-validation approach in the paper. Regarding the unsupervised methodologies, one can exploit the following alternative strategies:
>
> 1.  **Eigengap heuristic** [1]: The number $K$ is often found using the eigengap heuristic:
>
> $K = \text{argmax}\_{i}{(\lambda\_{i+1}-\lambda\_i)}$.
>
> A large gap indicates a natural cutoff point.
>
> 2. **Energy-based Criterion** [2]: The Laplacian EVD can be viewed like PCA, such that one can choose enough eigenvectors to explain a certain percentage of *spectral energy*. Therefore, the total spectral energy is defined as:
>
> $E\_{total}:=\sum\_{i=1}^{n}{\lambda\_i}$.
>
> One can thus choose the smallest $K$ such that $\frac{\sum_{i=1}^{K}{\lambda_i}}{E_{total}}\ge\eta$, where $\eta\in[0,1]$ is a threshold like 90\%.
>
> 3. **Spectral Gap Ratio or Knee Detection** [3]: Instead of just picking the largest gap, use a normalized eigengap ratio:
>
> $\text{gap ratio}(K):=\frac{\lambda_{K+1}-\lambda_{K}}{\lambda_{K}}$.
>
> This method accounts for the scaling of eigenvalues and can find relative jumps.
>
> 4. **Spectral Entropy-Information-Theoretic Criteria** [4]: The spectral entropy can be defined as:
>
> $H:=-\sum\_{i=1}^{n}{p\_i\log{p\_i}}; \text{ where } p\_i=\frac{\lambda\_i}{\sum\_{j}{\lambda\_j}}$.
>
> Therefore, one can choose $K$ where the entropy contribution of the next eigenvalues becomes negligible.
>
> Apart from introducing and listing the relevant methods, we have also experimentally applied these strategies on a subset of the $\texttt{high-school}$ node classification dataset in the following table, accompanied by the accuracy performance. Note the supervised methodology picked $\sim5.1\\%$ of eigenvalues for $\textbf{L}_1$, which is consistent with the unsupervised methods ($\sim5.5\\%$).
>
> **Table: Selected optimal values of eigenvalue-eigenvector $(K)$ pairs from $\sim$ 5800 ones for $\mathbf{L}_1$ by different relevant methods.**
> | |Eigengap | Energy-based | Knee detection | Spectral entropy|Cross-Validation|
> |---|---|---|---|---|---|
> |K|322 | 312 | 289 |312|300|
> |Acc|91.7 | 92.0 | 91.6 |92.0|91.9|
>
> Regarding the effect of varying $K$ on the performance, we provide comprehensive experimental results on a subset of the $\texttt{high-school}$ node classification dataset in the following table. From these results, we can conclude that choosing the best $K$ heavily relies on the specific data distribution, but generally, only a small fraction of the eigenvalues is enough to reach acceptable performance. We also observe that increasing $K$ does not necessarily lead to better performance, as high-frequency components might contain more noisy or non-relevant content [5].
>
> **Table: Performance (accuracy) results across a selected set of eigenvalue-eigenvector $(K)$ pairs from $\sim$ 300 ones for $\mathbf{L}_0$.**
> |$1\\%$| $3\\%$ | $10\\%$ | $30\\%$ | $50\\%$ | $70\\%$|
> |---|---|---|---|---|---|
> |88.0$\pm$5.2 | 90.6$\pm$2.1 | **93.4$\pm$3.4** | $91.4\pm1.9$ | 90.2$\pm$3.2 | 90.4$\pm$1.6|
>
> ---
> References:
>
> [1] J. Shi, et al, “Normalized cuts and image segmentation,” IEEE TPAMI, 2000
>
> [2] M. Belkin, et al, “Laplacian eigenmaps for dimensionality reduction and data representation,” Neural computation, 2003
>
> [3] I. M. Johnstone, “On the distribution of the largest eigenvalue in principal components analysis,” The Annals of statistics, 2001
>
> [4] M. De Domenico, et al, “Spectral entropies as information-theoretic tools for complex network comparison,” Physical Review X, 2016
>
> [5] Dong, X., et al. Learning Laplacian matrix in smooth graph signal representations. IEEE TSP 2016.

---

> > ### Comment · Reviewer_tWza · 2025-08-05
> >
> > I thank the authors for providing additional comments and experiments to address my concerns about the applicability of eigen-decomposition within the proposed pipeline. I would raise my score to accept.

---

### Official Review · Reviewer_yoZd · 2025-07-03

**Clarity:** 3
**Significance:** 3
**Originality:** 3
**Rating:** 5
**Confidence:** 3

**Summary:**

The work proposes a continuous simplicial neural networks (COSIMO). They use simplicial complexes to capture high-order geometrical relationships for a more sophisticated understanding of geometrical objects. In addition, they utilize the theory of differential equations to obtain the continuous formulation of the method to control the oversmoothing effect, which is a central problem in GNNs. They provided a mathematical evaluation of robustness against geometrical noise and oversmoothing. The experiments validate the mathematical theory and demonstrate that the proposed method has a competitive performance against SotA methods.

**Questions:**

What would be potential applications where higher-order relationships are essential?

**Ethical Concerns:**

["NO or VERY MINOR ethics concerns only"]

**Final Justification:**

The authors addressed all the concerns raised by the reviewer. Therefore, I would recommend acceptance.

**Limitations:**

yes

**Quality:**

3

**Strengths And Weaknesses:**

Strengths:
* The method is clearly explained and easy to follow.
* The authors provided the mathematical guarantees for stability and oversmoothing, which is essential to the community.

Weaknesses
* The strength to use higher-order relationships is not clear. In addition, the ideas for continuous evolution are not new. For instance, the community knows Neural ODE [Chen+ 2018] and GRAND [Chamberlain+ 2021]. Therefore, the authors should compare at least GRAND or its variants to show the effectiveness of higher-order relationship quantitatively.
* Evaluation of the tradeoff between oversmoothing and accuracy is not evaluated. It turned out that reducing the t (time) parameter in COSIMO suppresses oversmoothing, but it limits the lengths of interaction because it corresponds to short-time diffusion. Therefore, the reviewer recommends adding an evaluation of that tradeoff. Otherwise, if we do not care about the accuracy, it would be sufficient to stack only one GNN layer to suppress oversmoothing.
* Stability analysis (Section 6.3) needs comparison with other methods, because the readers cannot judge the robustness of the proposed method in the current form.

---

> ### Author Rebuttal · Authors · 2025-07-28
>
> We appreciate the reviewer for providing constructive comments on our paper. We'll include the feedback in the camera-ready paper. In the following, we have addressed the reviewer's concerns:
>
> **W1:** To complement the experimental analysis, we compare COSIMO against GRAND and Graph-coupled oscillator networks (GraphCON) [1] in the following table. We observe the superiority of COSIMO over GNNs, which only rely on the node space. This is a general observation in the comparison of SNNs vs. GNNs when higher-order information is available.
>
> |$\textbf{Method}$|$\texttt{high-school}$|$\texttt{senate-bills}$|
> |---|---|---|
> |GCN|0.40$\pm$0.04|$\underline{0.67\pm0.06}$|0.58$\pm$0.05|
> |GAT|0.34$\pm$0.05|0.50$\pm$0.04|0.57$\pm$0.06|
> |GraphSage|0.27$\pm$0.05|0.54$\pm$0.03|0.57$\pm$0.06|
> |GIN|0.18$\pm$0.04|0.53$\pm$0.04|0.57$\pm$0.06|
> |GRAND|0.16$\pm$0.07|0.52$\pm$0.10|
> |GraphCON|0.39$\pm$0.08|0.58$\pm$0.04|
> |SCCNN|$0.81\pm0.01$|0.62$\pm$0.05|0.61$\pm$0.03|
> |SAN|$\underline{0.86\pm0.04}$|0.53$\pm$0.09|0.64$\pm$0.05|
> |COSIMO|$\textbf{0.90$\pm$0.05}$|$\textbf{0.69$\pm$0.08}$|$\textbf{0.79$\pm$0.01}$|
>
> **W2:** We provide comprehensive results on the $\texttt{high-school}$ node classification dataset averaged over three random seeds in the following table regarding the tradeoff between over-smoothing and accuracy. We observe that the proposed COSIMO can maintain the performance even with an increasing number of layers up to 16 in terms of accuracy and Dirichlet energy. We also observe that COSIMO is more robust against over-smoothing than SCCNN, where extremely smoothness happens after $n_l=4$ and leads to severely reduced performance. The key observation and trend here are also fairly consistent with the results in Figure 3.
>
> **Acc:**
> |**Method** | $n_l$=1 | $n_l$=2 | $n_l$=4 | $n_l$=8 | $n_l$=16 | $n_l$=32|
> |---|---|---|---|---|---|---|
> |SCCNN | 57 | 81 | 43 | 16 | 12 | 10 |
> |COSIMO | 79.33 | 86.33 | 91.33 | 91.33 | 88.00 | 53.67|
>
> **Dirichlet Energy:**
> |**Method** | $n_l$=1 | $n_l$=2 | $n_l$=4 | $n_l$=8 | $n_l$=16 | $n_l$=32|
> |---|---|---|---|---|---|---|
> |SCCNN | 0.88| 0.80| 0.75| 0.53 | 0.42 | 0.07|
> |COSIMO | 0.94 | 0.92 | 0.92 | 0.92 | 0.88 | 0.19|
>
> **W3:** We have repeated the experiment in Section 6.3, including comparisons with the baseline methods SCCNN and SAN, and provided the results in the following table in terms of prediction error. We observe that the baselines are vulnerable in moderate to high amounts of noise (-5 or 0 db SNRs). In comparison, COSIMO has superior and more robust performance compared to the baselines even in the low-noisy regime (SNRs of 10 or 20 db).
>
> **SNR$_2$=-5:**
> |**Method** | SNR$_1$=-5 | SNR$_1$=0 | SNR$_1$=10 | SNR$_1$=20|
> |---|---|---|---|---|
> |SCCNN | 625.15 | 612.814 | 639.24 | 482.98|
> |SAN | 78281.68 | 32423.05 | 8984.80 | 4246.06|
> |COSIMO | 1.30 | 1.29 | 1.10 | 1.14|
>
> **SNR$_2$=0:**
> |**Method**| SNR$_1$=-5 | SNR$_1$=0 | SNR$_1$=10 | SNR$_1$=20|
> |---|---|---|---|---|
> |SCCNN | 191.45 | 57.84 | 46.44 | 49.54|
> |SAN | 526.45 | 1.65 | 1.00 | 1.09|
> |COSIMO | 1.34 | 1.16 | 1.03 | 0.92|
>
> **SNR$_2$=10:**
> |**Method** | SNR$_1$=-5 | SNR$_1$=0 | SNR$_1$=10 | SNR$_1$=20|
> |---|---|---|---|---|
> |SCCNN | 243.41 | 28.06 | 8.07 | 10.16|
> |SAN | 128.13 | 0.81 | 0.92 | 0.84|
> |COSIMO | 1.14 | 1.13 | 0.76 | 0.69|
>
> **Q1:** Based on the provided results regarding over-smoothing, stability, and performance, we argue that relying on simplicial complexes is meaningful when higher-order information is available. This is because SNNs, like COSIMO, can benefit from the higher-order dynamics. One practical example is the $\texttt{high-school}$ dataset, where friendship groups of multiple people (maximum cliques) are modeled as simplices. The description of the dataset has been explicitly mentioned on the official website as follows:
>
> "*This is a temporal higher-order network dataset, which here means a sequence of timestamped simplices where each simplex is a set of nodes. The dataset is constructed from interactions recorded by wearable sensors by people at a high school. The sensors record interactions at a resolution of 20 seconds (recording all interactions from the previous 20 seconds). Nodes are the people, and simplices are maximal cliques of interacting individuals from an interval.*"
>
> It is clear from our experimental results that relying on the typical graph structure (only node space) cannot leverage the information dynamics lying on the simplices in this dataset, so the performance of classical GNNs is not competitive against SNNs.
>
> To explore more diverse real-world applications from computer graphics to drug discovery and biotechnology, we kindly refer the reviewer to this recent ICML position paper [2].
>
> ---
> References:
>
> [1] T. K. Rusch, et al, “Graph-coupled oscillator networks,” ICML 2022
>
> [2] Papamarkou, et al. "Position: Topological deep learning is the new frontier for relational learning". ICML 2024.

---

> > ### Comment · Reviewer_yoZd · 2025-08-04
> >
> > Thank you for the detailed response and additional experiments. Regarding W2, if I understand correctly, the results are obtained by changing the number of layers with constant $t$, but how about the relationship between $t$ and accuracy? I am curious about this because the authors state that $t$ is the key parameter to control over-smoothing.

---

> > > ### Author Response · Authors · 2025-08-05
> > > **More clarification on W2:**
> > >
> > > We again thank the reviewer for providing constructive comments. We have addressed the remaining question as follows:
> > >
> > > Firstly, we should emphasize that as stated in Remark 5.6 in our paper, and due to the differentiability of our framework $t$ w.r.t. the simplicial receptive fields, in all the experimental results (except the over-smoothing analysis in Section 6.2), the values of $t$ are learned during the training process, *i.e.*, $t$ are parameters, not hyperparameters. We considered $t$ as a hyperparameter only in the over-smoothing analysis in Section 6.2 to illustrate the effect of different values for $t$ on the over-smoothing phenomenon.
> > >
> > > However, for further clarification and to fully address the reviewer's comment, we provide the node classification accuracy (Acc) and normalized Dirichlet Energy (DE) over the $\texttt{high-school}$ dataset across varying simplicial receptive field ($t$) in the following tables, *i.e.*, we treat $t$ as a hyperparameter. We observe that the Acc (and DE) is getting improved (best result) by increasing $t$ to $10^{-2}$, and then degrading by probably facing over-smoothing (especially by relying on DE getting generally smaller after this point). This shows that there is an optimal interval for tuning $t$ if it were to be a hyperparameter for getting good results, and not just necessarily by choosing a small $t$. This table also provides the results in the case where COSIMO learned $t$ during the training process. In this case, $\hat{t}=0.04$ was obtained after training, which is close to the best $t=10^{-2}$ when $t$ is treated as a hyperparameter. This shows COSIMO does not need to perform computationally demanding greedy hyperparameter optimization for $t$.
> > >
> > > **The node classification accuracy (Acc) and normalized Dirichlet Energy (DE) across varying simplicial receptive field ($t$):**
> > >
> > > |  | $t=10^{-4}$ | $t=10^{-3}$ | $t=10^{-2}$ | $t=10^{-1}$|$t=1$|$\hat{t}=0.04$|
> > > |---|---|---|---|---|---|---|
> > > |Acc | 87.9 | 89.4 | 89.4 | 71.2 | 69.7 |**90.4**|
> > > |DE| 0.842 | 0.849 | 0.849 | 0.824 | 0.831 |**0.850**|

---

> > > > ### Comment · Reviewer_yoZd · 2025-08-06
> > > >
> > > > Thank you for the response. Now everything is clear.
> > > >
> > > > Accordingly, I will increase my score to accept. Good work!

---

### Official Review · Reviewer_hz14 · 2025-07-05

**Clarity:** 2
**Significance:** 3
**Originality:** 3
**Rating:** 5
**Confidence:** 5

**Summary:**

The paper introduces COSIMO, a novel simplicial neural network rooted in Topological Deep Learning (TDL), designed to process data on simplicial complexes—mathematical structures that extend beyond graphs to model complex, higher-order relationships like nodes, edges, triangles, and beyond. TDL, as highlighted in the paper, leverages these topological structures to capture multi-way interactions, making it ideal for applications such as trajectory prediction and mesh processing. COSIMO advances TDL by shifting from the discrete, layer-based processing of traditional simplicial neural networks to a continuous framework, using partial differential equations based on Hodge Laplacians to model signal evolution across simplicial dimensions over time. This continuous approach enables COSIMO to dynamically handle interactions between lower-dimensional (e.g., edges) and higher-dimensional (e.g., triangles) structures, a core strength of TDL’s topological perspective.

COSIMO’s TDL framework incorporates independent diffusion processes for lower and upper simplicial structures and a coupled process to integrate them, enhancing expressivity for complex relational data. The paper provides rigorous theoretical analysis, proving COSIMO’s stability under simplicial perturbations and its ability to control over-smoothing—a challenge in TDL where deep models lose discriminative power—through adjustable receptive fields. Experimentally, COSIMO is evaluated on TDL-relevant tasks like ocean drift prediction, partial shape regression on the Shrec-16 dataset, and node/graph classification on social networks, showing superior performance compared to existing TDL methods, with balanced runtime and memory usage. The authors also explore parameter impacts, such as the number of branches, and confirm permutation equivariance, ensuring alignment with TDL’s emphasis on structural symmetries.

**Questions:**

1. Provide Detailed Experimental Specifications
Question/Suggestion: Could the authors provide a comprehensive description of the experimental setup, including specific hyperparameter values (e.g., learning rate, optimizer type, batch size), data splits, and training protocols, either in the main text or a dedicated appendix section? The current details in Section 6 and appendices (e.g., Appendix K) are incomplete, lacking specifics on how hyperparameters were chosen or tuned, which hinders reproducibility. I think since this one of the first papers : TDL+PDE, this will be very important for reproducibility.

Guidance for Action: Include a table or subsection listing all hyperparameters for each dataset (e.g., Shrec-16, ocean drift), specifying values for learning rate, optimizer (e.g., Adam, SGD), batch size, number of epochs, and any regularization techniques. Clarify the data split ratios (e.g., train/validation/test) and describe the hyperparameter tuning process (e.g., grid search, random search). If space is limited, a supplemental table or a pointer to the open-access code with these details would suffice.


2. Clarify Distinction from Graph Neural Diffusion Models
Question/Suggestion: The paper draws parallels with graph neural diffusion models [16, 37], but the unique contributions of COSIMO’s continuous PDE-based framework on simplicial complexes are not fully delineated. Could the authors explicitly clarify how COSIMO differs from or improves upon graph neural diffusion approaches, particularly in terms of architecture and theoretical advantages for TDL?

Guidance for Action: Add a subsection in Section 4 or the introduction comparing COSIMO’s PDE-based architecture (e.g., use of Hodge Laplacians for lower, upper, and coupled dynamics) to graph neural diffusion models (e.g., GRAND [37]). Highlight specific architectural differences (e.g., handling higher-order simplices vs. graph nodes/edges) and theoretical benefits (e.g., stability or expressivity for TDL tasks). A table contrasting key features (e.g., input structure, dynamics, computational complexity) or a paragraph discussing how COSIMO extends diffusion to simplicial complexes would be effective.


 3. Expand Discussion of Limitations and Failure Cases
Question/Suggestion: The discussion of limitations (e.g., computational complexity, over-squashing in Appendix O) is brief and mostly relegated to appendices. Could the authors expand this in the main text, detailing specific scenarios where COSIMO might underperform (e.g., sparse or noisy simplicial complexes) and proposing potential mitigations?

Guidance for Action: In the conclusion or a dedicated subsection, discuss at least two specific failure cases (e.g., performance on sparse simplicial complexes with few higher-order simplices, or sensitivity to noisy input signals). For each, explain why COSIMO might struggle (e.g., due to reliance on dense Hodge Laplacian computations) and suggest mitigations (e.g., sparse matrix techniques, noise-robust regularization). Reference empirical or theoretical evidence, such as sensitivity analyses (e.g., Figure 7) or related TDL work [54], to ground the discussion. Aim for 1–2 paragraphs in the main text, with additional details in appendices if needed.


 4. Provide Practical Guidance for Parameter Selection
Question/Suggestion: The paper shows how parameters like the receptive field (tau) and branch count (M) impact performance (e.g., Figure 7), but offers little guidance on selecting these for new TDL tasks. Could the authors provide practical recommendations for choosing these parameters, supported by empirical or theoretical insights?

Guidance for Action: Add a paragraph or subsection in Section 5 or the experiments (Section 6) outlining a strategy for selecting tau and M. For example, suggest starting values based on dataset characteristics (e.g., simplicial complex size, density of higher-order simplices) and adjusting based on validation performance. Reference Figure 7’s results (e.g., M=3 as a sweet spot for ocean drift) and theoretical insights (e.g., tau’s role in controlling over-smoothing from Theorem 5.3). If possible, provide a heuristic or algorithm (e.g., cross-validation over a range of tau values) and test it on one dataset to demonstrate efficacy.

**Ethical Concerns:**

["NO or VERY MINOR ethics concerns only"]

**Final Justification:**

All my concern have been thoroughly addressed by the authors. I increased my score accordingly as they did additional experiments and justified a few more claims theoretically.

**Limitations:**

Yes

**Quality:**

3

**Strengths And Weaknesses:**

Quality
This paper’s got some solid stuff going for it. The theoretical backbone is impressive—they’ve worked out proofs (like Proposition 4.1 and Theorem 5.3 in the appendices) that show COSIMO is stable under simplicial perturbations and can handle over-smoothing, leaning on math like Hodge Laplacians. It’s the kind of rigor that makes you feel they’ve thought this through. Experimentally, they test COSIMO on tasks like ocean drift prediction and shape regression on the Shrec-16 dataset, and Figure 5 shows it performs well against other TDL models while keeping runtime and memory in check. I like that they’ve shared their code and datasets, which is a big plus for reproducibility, especially since you’re into digging into results. Section 4.3’s breakdown of computational complexity, using eigenvalue decomposition for Hodge filters, shows they’re thinking about practical implementation, too.

On the other side, the experimental details could use more meat. They talk about setups in Section 6 and appendices, but stuff like exact hyperparameters or how they chose the optimizer is a bit fuzzy, which might make it tricky to replicate. The datasets are decent for TDL, but it’s mostly Shrec-16 and ocean drift—adding something like a biological network could show COSIMO’s got broader chops. Also, the eigenvalue decomposition they rely on can get heavy for big simplicial complexes. They mention this, but I wish they’d tossed in a few more ideas for making it less resource-hungry.

Clarity
The paper’s laid out nicely, with a clear path from the intro to preliminaries (Section 3), model details, and experiments. For someone familiar with TDL or graph neural networks, it’s pretty approachable—they explain simplicial complexes and Hodge Laplacians well enough, and Figure 1’s visual of a simplicial complex helps a lot. The theoretical bits, like how they tackle over-smoothing (Figures 3 and 6), are straightforward, with equations like the PDE-based signal evolution in Figure 2 giving a good sense of the architecture, which I know you care about. Figures like Figure 7, showing how tweaking the branch parameter affects results, make the practical side easy to grasp.

That said, the math-heavy parts can feel a bit thick if you’re not already familiar with TDL or algebraic topology. Terms like “Hodge Laplacians” or “permutation equivariance” get some context, but a simple example or two could make them less daunting for a wider audience, maybe not in the main text. Also, a lot of the good content, like proofs and dataset details, is stashed in the appendices, which is standard but leaves the main text a tad thin on quick takeaways. I had to switch back and forth more than I’d prefer. And while they show how parameters like the receptive field or branch count impact performance, they don’t give much guidance on picking those for new TDL problems, which could leave practitioners scratching their heads.

Significance
COSIMO feels like a meaningful step for TDL. Its continuous, PDE-based approach shakes up the usual discrete SNNs, making it easier to model dynamic interactions in simplicial complexes, which is super relevant as TDL gains traction. The applications they mention in Appendix O—like ocean navigation, social network analysis, and protein classification—hint at how TDL can tackle real-world challenges, which is exciting. Their focus on over-smoothing (Section 5) remains a standing problem in TDL and graph models, and they back it up with theory and results.

But I wonder if some might see it as a solid increment. It builds on existing ideas like graph neural diffusion and simplicial filters, so it’s not reinventing the wheel. The limitations section, covering things like computational costs or over-squashing, is there but feels a bit tacked on in the appendices. A deeper look at when COSIMO might not work well, like on sparse complexes, would make its impact clearer. Also, its focus on simplicial complexes might make it a niche player for the broader machine learning crowd, which could limit its splash outside TDL circles.

 Originality
The continuous framework is where COSIMO stands out. Using PDEs to model signal evolution on simplicial complexes is a fresh angle for TDL, moving past the fixed filters of traditional SNNs. Their use of Hodge Laplacians in a neural network setup is pretty creative, pulling from topological signal processing [19] but giving it a deep learning twist. The permutation equivariance proof in Appendix J is a nice touch, too, ensuring the model plays nicely with the symmetries of simplicial complexes, which isn’t always covered in TDL papers.

That said, it’s not entirely out of left field. COSIMO leans on concepts from TDL and graph neural networks, like diffusion models  and over-smoothing work , so it’s more of a smart remix than a totally new idea. The tasks they test—trajectory prediction, shape regression—are standard TDL fare, so there’s no big surprise in the applications. A novel TDL use case could’ve pushed the originality further. Also, the overlap with graph neural diffusion might make some people wonder how much is truly new versus adapting those ideas to simplicial complexes.

---

> ### Author Rebuttal · Authors · 2025-07-28
>
> We appreciate the reviewer for providing constructive comments on our paper. We'll include the feedback in the camera-ready paper. In the following, we have addressed the reviewer's concerns:
>
> **Q1**: Firstly, we clarify that we used cross-validation for tuning the hyperparameter. For experimental results on $\texttt{synthetic}$ and $\texttt{ocean-drifts}$, we followed the experimental settings from reference papers [1,2]. In addition to adding a comprehensive table of details like the following table for the camera-ready, we will publicly release the codes (which have already been uploaded alongside our submission) and refer to them to make these settings as clear as possible.
>
> |$\textbf{Hyperparam}$|$\texttt{synthetic}$|$\texttt{ocean-drifts}$|$\texttt{Shrec-small}$|$\texttt{Shrec-full}$|$\texttt{high-school}$|$\texttt{senate-bills}$|$\texttt{proteins}$|
> |---|---|---|---|---|---|---|---|
> |$lr$|$5\times10^{-3}$|$5\times10^{-2}$|$10^{-2}$|$10^{-2}$|$10^{-3}$|$10^{-2}$|$10^{-3}$|
> |Optimizer|ADAM|ADAM|ADAM|ADAM|ADAM|ADAM|ADAM|
> |Batch size|100|100|256|512|256|256|256|
> |$n_\text{epochs}$|1000|1000|100|100|700|100|30|
> |$M$|3|3|1|1|1|1|1|
>
> **Q2:** In order to make the connection between the proposed COSIMO and reference Graph-PDE counterparts [3,4,10], we first highlight the generalization aspect of simplicial complexes over graphs. For instance, let's consider a simple case of a simplicial complex containing nodes, edges, and triangles. In this case, by relying on graph structure, we neglect edge and triangle dynamics. Therefore, we only deal with $\mathbf{L}_0$ (in the nodes space with $k=0$), and equations (6) and (7) in our framework are canceled. On the other hand, since $k=0$, only the lower node space is present and equation (5) in our framework turns to $\frac{\partial \mathbf{x}(t)}{\partial t}=-\mathbf{L}_0\mathbf{x}(t)$. Then, by assuming the usage of the normalized version of this Laplacian, *i.e.*, $\hat{\mathbf{L}}_0=\mathbf{I}-\mathbf{A}$ with $\mathbf{A}$ being the normalized adjacency matrix, this equation turns to $\frac{\partial \mathbf{x}(t)}{\partial t}=(\mathbf{A}-\mathbf{I})\mathbf{x}(t)$, which is equivalent to eq. (2) in GRAND [4] and eq. (7) in CGNN [3] frameworks in the isotropic linear cases. Therefore, COSIMO generalizes the CGNN and GRAND models by considering edge and triangular dynamics. Apart from these theoretical/architectural differences, we have also experimentally compared COSIMO against GRAND [4] and Graph-coupled oscillator networks (GraphCON) [10] in the following table, showcasing the superior performance by relying on higher-order dynamics.
>
> |$\textbf{Method}$|$\texttt{high-school}$|$\texttt{senate-bills}$|
> |---|---|---|
> |GCN|0.40$\pm$0.04|$\underline{0.67\pm0.06}$|0.58$\pm$0.05|
> |GAT|0.34$\pm$0.05|0.50$\pm$0.04|0.57$\pm$0.06|
> |GraphSage|0.27$\pm$0.05|0.54$\pm$0.03|0.57$\pm$0.06|
> |GIN|0.18$\pm$0.04|0.53$\pm$0.04|0.57$\pm$0.06|
> |GRAND|0.16$\pm$0.07|0.52$\pm$0.10|
> |GraphCON|0.39$\pm$0.08|0.58$\pm$0.04|
> |SCCNN|$0.81\pm0.01$|0.62$\pm$0.05|0.61$\pm$0.03|
> |SAN|$\underline{0.86\pm0.04}$|0.53$\pm$0.09|0.64$\pm$0.05|
> |COSIMO|$\textbf{0.90$\pm$0.05}$|$\textbf{0.69$\pm$0.08}$|$\textbf{0.79$\pm$0.01}$|
>
> **Q3:** Apart from the outlined possible limitations in the conclusion section, we also point out the case of *i)* eventually facing over-smoothing in Figure 3, and *ii)* an extremely noisy regime (for example, $SNR_1=-5$ and $SNR_2=-5$ in Figure 4). To mitigate these limitations in these cases, it has been shown that equipping the underlying set of PDEs with **source terms** [5] can effectively improve the stability and robustness to over-smoothing. Therefore, the corresponding equations of our framework, *i.e.*, independent and joint dynamics, turn to:
>
> $\frac{\partial \mathbf{x}\_{k,d}(t\_d)}{\partial t\_d} = -\mathbf{L}\_{k,d}\mathbf{x}\_{k,d}(t\_d)\rightarrow\frac{\partial \mathbf{x}\_{k,d}(t\_d)}{\partial t\_d} = -\mathbf{L}\_{k,d}\mathbf{x}\_{k,d}(t\_d)+\mathbf{E}\_d,$
>
> $\frac{\partial \mathbf{x}\_{k,u}(t\_u)}{\partial t\_u} = -\mathbf{L}\_{k,u}\mathbf{x}\_{k,u}(t\_u)\rightarrow\frac{\partial \mathbf{x}\_{k,u}(t\_u)}{\partial t\_u} = -\mathbf{L}\_{k,u}\mathbf{x}\_{k,u}(t\_u)+\mathbf{E}\_u,$
>
> $ \frac{\partial \mathbf{x}\_{k}(t\_d, t\_u)}{\partial t\_d} + \frac{\partial \mathbf{x}\_{k}(t\_d, t\_u)}{\partial t\_u} = -\mathbf{L}\_{k,d}\mathbf{x}\_{k}(t\_d, \infty) - \mathbf{L}\_{k,u}\mathbf{x}\_{k}(\infty, t\_u)\rightarrow\frac{\partial \mathbf{x}\_{k}(t\_d, t\_u)}{\partial t\_d} + \frac{\partial \mathbf{x}\_{k}(t\_d, t\_u)}{\partial t\_u} = -\mathbf{L}\_{k,d}\mathbf{x}\_{k}(t\_d, \infty) - \mathbf{L}\_{k,u}\mathbf{x}\_{k}(\infty, t\_u)+\mathbf{E}\_{d,u},$
>
> where $\mathbf{E}\_d$, $\mathbf{E}\_u$, and $\mathbf{E}\_{d,u}$ are source terms. Although this modification might bring the above-mentioned advantages, the solutions pose more computational burden, as detailed in different frameworks in [5].
>
> **Q4:** As stated in Remark 5.6, and due to the differentiability of our framework *w.r.t.* the simplicial receptive fields, in all the experimental results (except the over-smoothing analysis in Section 6.2), the values of $t$ are learned during the training process, *i.e.*, $t$ are parameters, not hyperparameters. We considered $t$ as hyperparameters only in the over-smoothing analysis in Section 6.2 to illustrate the effect of different values for $t$ on the over-smoothing phenomenon. In general, we use cross-validation over a range of possible values to tune the hyperparameters of COSIMO. Regarding the hyperparameter $M$, we also exploited a multi-branch architecture in trajectory prediction tasks in which we have already provided its sensitivity and accuracy trade-off in the Appendix, *i.e.*, Figure 7, by using a cross-validation scheme on the corresponding range for $M$. Furthermore, one can select a proper $K$ using supervised (cross-validation) or unsupervised methodologies. We used the cross-validation approach in the paper. Regarding the unsupervised methodologies, one can exploit the following alternative strategies:
>
> 1.  **Eigengap heuristic** [6]: The number $K$ is often found using the eigengap heuristic:
>
> $K = \text{argmax}\_{i}{(\lambda\_{i+1}-\lambda\_i)}$.
>
> A large gap indicates a natural cutoff point.
>
> 2. **Energy-based Criterion** [7]: The Laplacian EVD can be viewed like PCA, such that one can choose enough eigenvectors to explain a certain percentage of *spectral energy*. Therefore, the total spectral energy is defined as:
>
> $E\_{total}:=\sum\_{i=1}^{n}{\lambda\_i}$.
>
> One can thus choose the smallest $K$ such that $\frac{\sum_{i=1}^{K}{\lambda_i}}{E_{total}}\ge\eta$, where $\eta\in[0,1]$ is a threshold like 90\%.
>
> 3. **Spectral Gap Ratio or Knee Detection** [8]: Instead of just picking the largest gap, use a normalized eigengap ratio:
>
> $\text{gap ratio}(K):=\frac{\lambda_{K+1}-\lambda_{K}}{\lambda_{K}}$.
>
> This method accounts for the scaling of eigenvalues and can find relative jumps.
>
> 4. **Spectral Entropy-Information-Theoretic Criteria** [9]: The spectral entropy can be defined as:
>
> $H:=-\sum\_{i=1}^{n}{p\_i\log{p\_i}}; \text{ where } p\_i=\frac{\lambda\_i}{\sum\_{j}{\lambda\_j}}$.
>
> Therefore, one can choose $K$ where the entropy contribution of the next eigenvalues becomes negligible.
>
> Apart from introducing and listing the relevant methods, we have also experimentally applied these strategies on a subset of the $\texttt{high-school}$ node classification dataset in the following table, accompanied by the accuracy performance. Note the supervised methodology picked $\sim5.1\\%$ of eigenvalues for $\textbf{L}_1$, which is consistent with the unsupervised methods ($\sim5.5\\%$).
>
> **Table: Selected optimal values of eigenvalue-eigenvector $(K)$ pairs from $\sim$ 5800 ones for $\mathbf{L}_1$ by different relevant methods.**
> | |Eigengap | Energy-based | Knee detection | Spectral entropy|Cross-Validation|
> |---|---|---|---|---|---|
> |K|322 | 312 | 289 |312|300|
> |Acc|91.7 | 92.0 | 91.6 |92.0|91.9|
>
> In conclusion and by observing results on various datasets, choosing 5-10$\\%$ of eigenvalue-eigenvector pairs and number of branches ($M$) from 1-3 seems to provide a fair compromise of performance on downstream tasks.
>
> ---
> References:
>
> [1] M. Roddenberry, et al, “Principled simplicial neural networks for trajectory prediction,” ICML 2021
>
> [2] M. Yang, et al, “Convolutional learning on simplicial complexes,” arXiv 2025
>
> [3] L.-P. Xhonneux, et al, “Continuous graph neural networks,” ICML 2020
>
> [4] B. Chamberlain, et al, “Grand: Graph neural diffusion,” ICML 2021
>
> [5] A. Han, et al, “From continuous dynamics to graph neural networks: Neural diffusion and beyond,” TMLR 2024
>
> [6] J. Shi, et al, “Normalized cuts and image segmentation,” IEEE TPAMI, 2000
>
> [7] M. Belkin, et al, “Laplacian eigenmaps for dimensionality reduction and data representation,” Neural computation, 2003
>
> [8] I. M. Johnstone, “On the distribution of the largest eigenvalue in principal components analysis,” The Annals of statistics, 2001
>
> [9] M. De Domenico, et al, “Spectral entropies as information-theoretic tools for complex network comparison,” Physical Review X, 2016
>
> [10] T. K. Rusch, et al, “Graph-coupled oscillator networks,” ICML 2022

---

> > ### Comment · Reviewer_hz14 · 2025-08-09
> > **All comments have been been addressed bt the authors**
> >
> > Thanks for the authors for taking the time to address all my comments. I have read the other responses that the authors made to my comments to other reviewers and I find them satisfactory.  I will increase my score accordingly.

---

### Author Response · Authors · 2025-08-08
**We sincerely thank the reviewers**

We sincerely thank the reviewers for their thorough assessments and constructive feedback, which have helped us clarify and strengthen our work. We greatly appreciate your engagement with our paper and the recognition of its contributions. Your detailed comments have been invaluable during this submission process, and we are grateful for your service to the community. Please let us know if there are any remaining questions or points we can further clarify.

Best regards,

Authors

---

### Decision · Program_Chairs · 2025-09-17

**Decision:**

Accept (poster)

**Comment:**

This paper proposes COSIMO, a novel simplicial neural network to process data on simplicial complexes. The method design seems effective and novel, and the paper demonstrates that their approach works well empirically and provides rigorous theoretical analysis. Following the reviewers' feedback, the authors can refine the paper to be more clear.

With positive scores from all reviewers, I vote for acceptance.